# The effect of motivational and instructional self-talk on attentional control under noise distraction

Liu Yang[1,2], Yingchun Wang[2]*

1 School of Psychology, Shanghai University of Sport, Shanghai, China, 2 School of Psychology, Beijing Sport University, Beijing, China

* 13521531556@163.com

## Abstract

Inhibition is the key factor of attentional control (AC). Basketball players are typically exposed to noise from the audience or opposing teams while competing. These distractions disrupt the attentional systems, ultimately compromise the athletes' inhibition ability and directly affect their performance on the court. Hence, effective AC strategies are crucial. Two studies were demonstrated to investigate the effects of noise distractions on attentional control and the moderating effect of self-talk. In Study 1, 36 participants undertook the Stroop task, showing an increased error rate with noise distraction. Thirty-nine national second-level basketball players participated in Study 2, where they engaged in the Antisaccade task under both quiet and noise-distraction conditions, employing different self-talk strategies. Results showed that instructional self-talk reduced the antisaccade error rate in quiet conditions, while motivational self-talk increased the error rate under noise distractions. These findings suggests that noise distraction reduces AC. In competition scenarios, basketball players are required to appropriately implement self-talk strategies to improve AC and prevent potential counterproductive effects.

## Introduction

Attentional control theory is based on the processing efficiency theory [1]. According to the theory, anxiety affects attentional control (AC) through disrupting the balance between goal-directed and stimulus-directed attentional systems [2]. Furthermore, anxiety escalates the allocation of attention to irrelevant stimuli, both internal and external, leading to bottom-up processing. This process affects key functions of AC (i.e., inhibition and shifting), especially inhibition, the primary indicator [1, 2]. Inhibition is a process that involves suppressing, blocking, or delaying responses to irrelevant stimuli in order to prevent attentional resources from being allocated to task-irrelevant stimuli [3].

AC and sports performance are closely related, and the quiet eye is one of the indicators of AC. The participants who have undergone quiet eye training have a higher percentage of basketball free throws [4]. Three-point shot performance under pressure is positively correlated

**Data Availability Statement:** All relevant data are within the manuscript and its Supporting Information files.

**Funding:** The author(s) received no specific funding for this work.

**Competing interests:** The authors have declared that no competing interests exist.

with quiet eye duration, suggesting that AC may fulfill an online control function [5]. Furthermore, experts have longer quiet eye duration and performance better on sport decision-making tasks than novices, causing experts could pay more attention to task-related cues and ignore the task-irrelated information [6].

After more than 20 years of development, however, attentional control theory has gradually expanded to cover other variables rather than being limited to the effects of anxiety on attentional control (e.g., noise distraction, cognitive load, ego depletion, working memory, etc.) [7–12]. Trimmel and Poelzl (2006) demonstrated that noise caused a decrease in AC and information processing in the cortical layer [13]. Trimmel et al. (2012) conducted further research and discovered that that noise from neighbors and aircraft both impaired learning ability when compared to the control group, with the potential mechanism being that noise affects attentional control and cognitive strategies during learning [14]. A meta-analysis of 242 research indicated that noise had a moderate negative influence on cognitive functions [15]. The mean effect size across the 191 cognitive tasks was $d$ = -0.34 (95%CI = -0.42 to -0.25), with the highest effect size for high loudness, short duration, $d$ = -0.68, (95%CI = -0.96 to -0.40), suggesting that short, loud noises have the most negative effect on AC [15].

Attentional focus in sport can be divided between external or internal dimensions. External focus cues direct a player's attention to the environment or the effects of their movements on the environment, while internal focus cues direct a player's attention to their own body movements or muscle engagement [16]. In basketball, players must be aware of their surroundings and the movements of their teammates and opponents to make quick decisions, such as passing, shooting, or defending on the court [17]. However, focusing on misleading information, such as fake moves or trash talk from the opponent, can often result in incorrect decision making and incoordinate movements, ultimately affecting the performance [18, 19].

External distractions such as noise or light can potentially affect motor learners and hinder motor performance [20]. Basketball players usually compete under the noise from the audience or opponents and they are expected to finish the game in that noisy environment. The noise level on the field is frequently high, for instance, in American football, peak noise levels from the audience can reach 123-140dB, which makes it difficult for athletes to communicate on the field and puts the athlete's AC to the test [21]. Despite the athletes receiving 6 weeks of shooting training, there was a significant decrease in shooting performance in the presence of noise distraction compared to baseline levels (no noise distraction) [22]. This implies that psychological skills are necessary in order to counteract the negative effects of noise distraction and that skill training alone may not be sufficient.

One of the most frequently employed mental skills in sports is self-talk, which takes many different forms. There is empirical evidence that self-talk strategies are effective in enhancing sport/task performance through systematic [23] and meta-analytic reviews [24]. Self-talk can be categorized into motivational self-talk (e.g., "I can do it") and instructional self-talk (e.g., "elbow, wrist, shoot") based on its strategic characteristics [25]. Diverse self-talk cues engender distinct functional and performance outcomes. Specifically, both attentional and anxiety control cues have been observed to reduce the frequency of interfering thoughts while concurrently increasing confidence and effort levels [26]. Importantly, the efficacy of anxiety control cues in modulating anxiety control is distinctly more significant [26]. According to the valence of self-talk, it could be divided into positive self-talk (e.g., "good job") and negative self-talk (e.g., "I'm going to lose") [27]. Motivational self-talk seems more appropriate in competitive settings, whereas instructional self-talk seems more appropriate in training settings [28]. Instructional self-talk, primarily aimed at enhancing concentration, proved more beneficial for novel tasks by directing attention to skill outcomes [29]. However, for learned tasks, motivational self-talk demonstrated greater effectiveness over instructional self-talk [29]. In the

Dart-throwing task, motivational group improved more from the baseline to the final task than the performance of the control group, but there was no significant difference in performance change from the baseline to the final task between the instructional and control groups [30]. The effectiveness of self-talk hinges on its ability to shape one's appraisal of a task as a challenge or threat. Motivational self-talk serves to amplify an individual's perception of tasks as challenges, subsequently enhancing performance outcomes. Conversely, instructional self-talk, though providing procedural clarity and guidance, may not sufficiently recalibrate an individual's appraisal from perceiving a task as a threat to recognizing it as a challenge, thus the performance was not improved [30]. Theodorakis et al.(2000) and Hatzigeorgiadis et al. (2011) suggest that compare to motivational self-talk, instructional self-talk is more effective in fine motor control, whereas motivational self-talk is superior in strength and endurance sports when compared to instructional self-talk [24, 25]. Basketball players had better free throw performance and lower movement coordination variability when using instructional self-talk, which meant that they executed their movements with more consistency and fluidity, but motivational self-talk did not have a significant effect on either of these outcomes [31].

The empirical and theoretical research on why self-talk can influence exercise performance is still being developed and refined. The model proposed by Hardy et al. (2009) suggests that self-talk influences motor performance primarily through four factors: (1) cognition (e.g., attention); (2) motivation (e.g., self-confidence, self-efficacy); (3) behavior (e.g., motor skill); and (4) affect (e.g., anxiety) [32]. Another model proposed by Galanis et al. (2016) in a systematic review placed the focus on the mechanisms by which self-talk affects motor performance on attention and motivation, which were identified as two important variables in how self-talk affects motor performance [33]. In addition, a more insightful exploration suggests that frontal lobe generated decisions and actions are less connected to other perceptual information and top-down processing dominates the attentional system when participants use instructional self-talk [34]. In ego depletion state, self-talk could also improve attention functions (e.g., selective attention), which may be attributed to self-talk can counteract the effects of distracting stimuli or ego depletion [35, 36]. In summary, it can be concluded that self-talk affects performance by AC, and the correct use of self-talk facilitates the dominance of the goal-directed attention system, leading to AC on the task at hand.

In conclusion, attention may be affected by self-talk and noise distraction. Whereas there is little research reporting the relationship between noise distraction, self-talk, and AC. Hence, the present study aims to uncover the relationship via two experiments. AC will be assessed using the Stroop task and Antisaccade task, two well-established paradigms for investigating AC [8, 37, 38]. Study 1 will measure the effect of noise distraction on AC via the Stroop task, and the Study 2 will further examine the effects of noise distraction on basketball players and the moderating effect of self-talk via the Antisaccade task. Based on the previous findings, we hypothesized that (1) participants exhibit lower AC levels under noise distraction in the Stroop task and Antisaccade task; (2) Self-talk affects AC and moderates the effect of noise distractions on AC. Specifically, with noise distractions, only instructional self-talk group improved AC compared to motivational self-talk and control groups.

## Study 1

### Method

**Participants.** The sample size was computed by GPower 3.1 [39], by selecting "t -tests: Difference between two independent means (two groups)" as statistical test, tail (s) = two, effect size dz = 0.50, α err prob = 0.05, power (1-β err prob) = 0.80. Based on these parameters, a total of 34 participants were calculated to be necessary for the study.

Thirty-six university students (25 females, age = 22.00 ± 1.66, Mean ± SD) from Beijing Sport University, none of whom had any prior basketball training, participated in Study 1. All participants had normal or corrected-to-normal visual acuity. The study was approved by the Sports Science Experiment Ethics Committee of Beijing Sport University (ethical identification number: 2022107H). All participants gave written informed consent that they took part in the study voluntarily.

**Experimental design.** Single factor (condition: noise distraction condition, quiet condition) within-subjects design was used. The dependent variables were reaction time and error rate in the incongruent condition of the Stroop task [37].

**Experimental materials.** *Stroop task.* The Stroop task (Fig 1) was programmed by Psychopy 3.0 and was presented using a computer monitor (15", 1920 × 1080 resolution, frequency 60Hz). The task was comprised of the following three colors: red, blue, and green. Two trial types were included in this task: (1) incongruent trials, in which the word named a color incongruent with the ink color in which it was printed (e.g., the word "RED" in green color); (2) congruent trials, in which the word named a color congruent with the ink color (e.g., the word "RED" in red color). The ratio of congruent trials to incongruent trials was 1:2. Participants were instructed to ignore the word meaning of the stimulus and to use the keypad to determine the color of that word, pressing "1" for red, "2" for green, and "3" for blue.

In inconsistent condition, shorter reaction times and smaller error rates both reflect stronger inhibition.

*Noise distraction.* The noise distraction material comprises basketball commentary, the sound of basketball dribbling, whistles from the audience, and recorded voices [22]. The duration of the noise material is 4 minutes and 58 seconds. It is designed to be played continuously while the participant is performing the task. If the task duration exceeds 4 minutes and 58

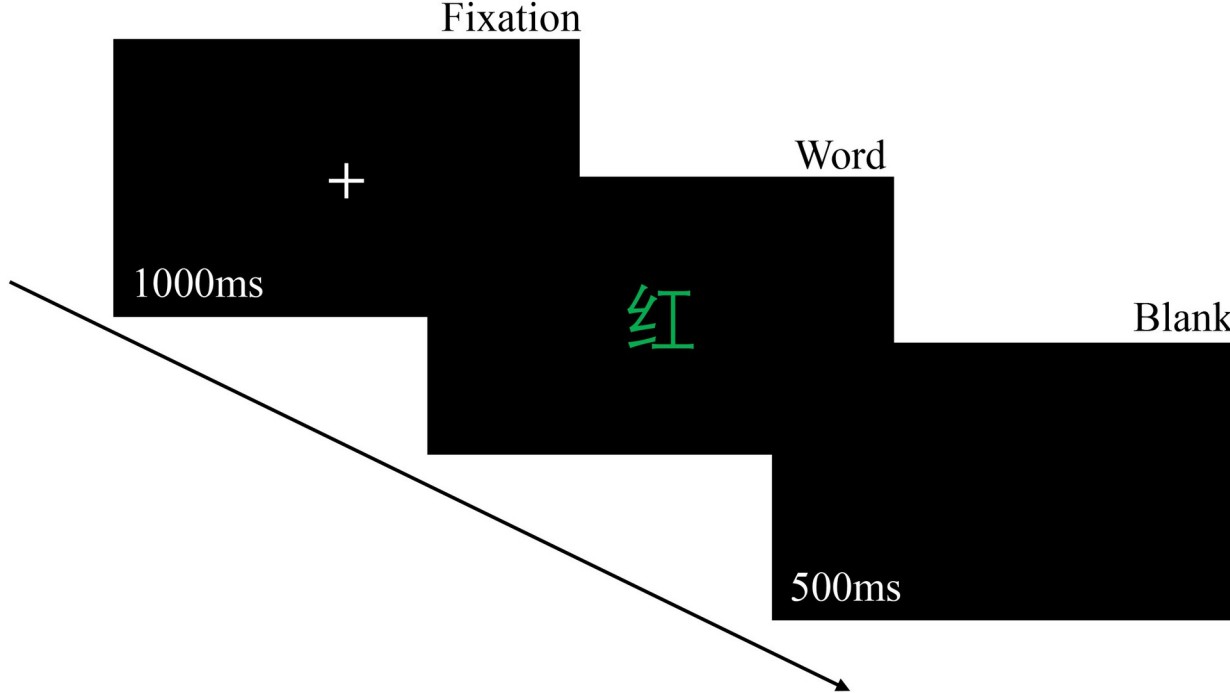

**Fig 1. Stroop task.** After 1000ms of fixation presentation, the stimulus ('红' means red in Chinese) appears until the participant responds and then disappears. Then 500ms blank screen appears.

seconds, the noise material should be replayed from the beginning and continue until the task is completed. The basketball commentary, excerpted from a particular game segment, spans a duration of 4 minutes and 58 seconds without any interruption. It maintains a moderate volume level and is presented in Chinese. The sound of basketball dribbling occurs at a frequency of once every second, with the volume set to the minimum level. The whistles from the audience manifest intermittently, with intervals ranging from 30 to 50 seconds, and are presented at the highest volume level. The recorded voices are comprised of "red, green, blue, up, down, right, miss the shot" in Chinese. They play uninterruptedly for a duration of 4 minutes and 58 seconds. The volume level is set to be lower than the basketball commentary but higher than the sound of basketball dribbling. The audio material was played on AirPods Pro in transparency mode. The playback intensity was monitored using the accessibility function provided by Apple's iOS operating system, with the actual playback intensity ranging from 85 to 95dB. [15].

**Procedure.**   After reading and signing the informed consent, participants completed 20 practice trials of Stroop task, four blocks of the formal experiment were then started and every block contains 45 trials. Two blocks are quiet conditions and two blocks are noise distraction conditions. The experimental sequence is based on the ABBA counterbalance design, where a random number either 1 or 2 determines the condition of the first block: if the number is 1 then it is quiet condition; if the number is 2 then it is noise distraction.

**Data analysis.**   Using SPSS 26.0, conducted paired sample $t$-test for error rate and reaction time of incongruent trials. Trials with a Z-score more than 2 or less than -2 for each participant were discarded.

## Result

As shown in Table 1, compare to quiet condition, the error rate was significantly higher under noise distraction condition ($p < .01$), however, there was no significant difference in reaction time ($p > .05$).

## Study 2

### Method

**Participants.**   The sample size was computed by GPower 3.1 [39], by selecting "ANOVA: Repeated measure, within-between interaction" as statistical test, effect size f = 0.25, α err prob = 0.05, power (1-β err prob) = 0.80, number of groups = 3, number of measurements = 2, and keeping the rest of the parameters default. Based on these parameters, a total of 42 participants were calculated to be necessary for the study.

Forty-two participants (all males) who come from Beijing Sport University participated in Study 2, all participants were national second-level basketball players and had normal or corrected-to-normal visual acuity. Two participants were excluded due to the qualified rate of less than 60% on the antisaccade trials and 1 person was excluded due to the key press error rate of more than 20%. Thirty-nine participants (age = 21.54 ± 2.49, Mean ± SD) were eventually included in Study 2. They have played basketball for 8.46 ± 2.34 (Mean ± SD) years and trained

**Table 1. Result of the Stroop task.**

| Variables | NDC ($N = 36$) | QC ($N = 36$) | $t$ | $p$ | Cohen's $d$ |
|---|---|---|---|---|---|
| Error rate | 7.60 ± 6.79 | 5.49 ± 4.69 | 2.82 | < .01 | 0.47 |
| RT (ms) | 932.72 ± 120.28 | 933.25 ± 125.05 | -0.04 | .97 | < -0.01 |

NDC: noise distraction condition; QC: quiet condition; RT: reaction time.

for 10.00 ± 5.99 (Mean ± SD) hours per week. The study was approved by the Sports Science Experiment Ethics Committee of Beijing Sport University (ethical identification number: 2022107H). All participants gave a written informed consent that they took part in the study voluntarily.

**Experimental design.** Two (condition: noise distraction condition, quiet condition) × three (group: motivational self-talk group, instructional self-talk group, control group) mixed experimental design was used. The condition is within variable and the group is between variable. The dependent variables were the latency of the first correct antisaccade, the error rate of the first saccade, and pupil diameter.

**Experimental materials.** *Antisaccade task.* The Antisaccade task was programmed by Ergolab 3.16 and presented using a computer monitor (24", 1920 × 1080 resolution, frequency 60Hz). The Tobii Pro Spectrum was used to record the participants' eye movements, with a sampling frequency of 1200Hz and an accuracy of up to 0.3˚. With five calibration points, video-based pupil- and corneal reflection eye tracking was used to capture the participants' eye movements.

In each trial, the participant starts by fixating on the Chinese word "准备" (i.e., "ready" in Chinese). Once this word vanishes, a cross fixation point appears for one second. A rectangle, serving as a distractor stimulus, then appears on either the left or right side of the screen. Participants are instructed to swiftly shift their gaze in the opposite direction of the distractor stimulus. After the distractor disappears, an arrow briefly shows for 200ms before being covered by a circular stimulus. This arrow, which randomly points left, right, or up, requires the participant to respond correctly to its direction (e.g., pressing the left button for a left-pointing arrow). As soon as the participant responds to the arrow's direction, the circular stimulus vanishes (refer to Fig 2 for details).

*Noise distraction.* Same as Study 1.

*Self-talk intervention.* Introduced the concept and advantages of self-talk at first. Then asked participant to use self-talk cue when 'ready' appears in every trial. Motivational self-talk group used 'I can' or 'I can do it' which are commonly used in previously research [31, 40]. There are no references about the instructional self-talk cue on the antisaccade task. However, according to Hardy (2006), the performance of relatively fine motor tasks requiring skill, timing, and accuracy was enhanced to a greater extent by instructional self-talk, focusing on the technical aspects of performance, than motivational self-talk. Therefore, 'look at the opposite' is used for instructional self-talk cue. It intended to remind participants to look in the opposite direction of the target, which is the goal and technical aspect of the task [27]. Control group talked nothing to themselves during whole experiment.

*Manipulation check.* Five-point Likert scale (1 = never, 5 = always) which contained 2 items was used: (1) I just said to myself something like 'I can' or 'I can do it'; (2) I just said to myself something like 'look at the opposite'.

**Procedure.** After reading and signing the informed consent, participants were randomly grouped and completed 20 practice trials of Antisaccade task, if the error rate of the first saccade in practice were higher than 20%, a new 20-trial was required, until the error rate was less than 20%. Four blocks of formal experiment were then started and every block contains 36 trials. The condition setting and sequence setting as in Study 1. After finishing the experiment, participants completed the manipulation check, some simple questions about whether they used self-talk cues and demographic information.

**Data analysis.** Using SPSS 26.0, conducted ANOVA for manipulation check and demographic information, 2 × 3 repeated measures analysis of variance (RMANOVA) with Group (control, motivational self-talk, instructional self-talk) as the between-subjects factor and Condition (noise distraction condition, quiet condition) as the within-subjects factor. Similar to

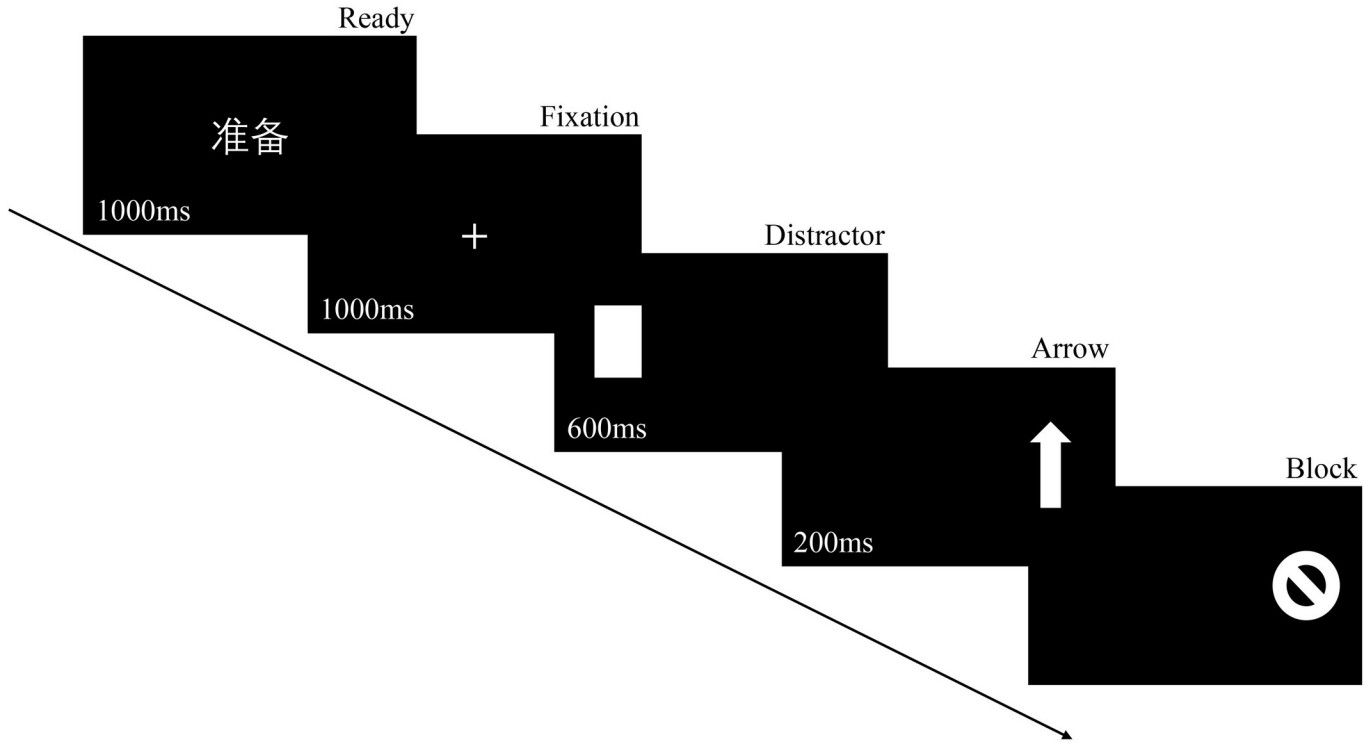

**Fig 2. Antisaccade task.** A 1000ms "ready" was presented first, followed by a 1000ms cross fixation, which the participant needed to look at before the cross fixation disappeared, otherwise it was considered an invalid trial. The distractor stimulus was then presented for 600ms, followed by a 200ms arrow after the distractor stimulus disappeared. After the arrow vanishes, a circular stimulus appears and remains until the participant responds to the arrow's direction.

previous study [38], the saccade threshold criterion was set at 30º /s and amplitude was set at 3º. The pupil diameter was the average diameter in one second from distractor stimulus appears. Anticipatory saccades with latencies less than 80ms or late saccades more than 600ms were discarded.

## Result

**Demographic information.** Descriptive statistics are provided in Table 2. The ANOVA analysis revealed a non-significant difference in the number of years playing basketball between the control, motivational self-talk and instructional self-talk groups ($F_{(2, 36)} = 1.21$, $p = .310$); and a non-significant difference in the number of hours of training per week ($F_{(2, 36)} = 1.03$, $p = .366$).

**Table 2. Demographic information and manipulation check.**

| Group | NOP | PBY $M$ $(SD)$ | THPW $M$ $(SD)$ | Item 1 $M$ $(SD)$ | Item 2 $M$ $(SD)$ |
|---|---|---|---|---|---|
| CG | 13 | 8.69 (4.52) | 9.62 (2.63) | 1.38 (0.65) | 1.92 (1.26) |
| MSTG | 13 | 7.08 (3.52) | 11.85 (9.54) | 4.23 (0.73) | 1.54 (0.97) |
| ISTG | 13 | 9.62 (4.52) | 8.54 (3.07) | 1.46 (0.66) | 4.23 (1.01) |
| Sum | 39 | 8.46 (4.24) | 10 (5.99) | 2.36 (1.50) | 2.56 (1.60) |

NOP: number of participants; PBY: playing basketball years; THPW: training hours per week; Item 1: I just said to myself something like 'I can' or 'I can do it'; Item 2: I just said to myself something like 'look at the opposite'; CG: control group; MSTG: motivational self-talk group; ISTG: instructional self-talk group.

**Manipulation check.** Descriptive statistics are provided in Table 2. The assumptions for conducting an ANOVA analysis were met. The ANOVA analysis revealed that motivational self-talk group had a higher score of the item that I just said to myself something like 'I can' or 'I can do it' compare to other groups ($F$ (2, 36) = 74.06, $p < .001$); and instructional self-talk group had a higher score of the item that I just said to myself something like 'look at the opposite' compare other groups ($F$ (2, 36) = 23.37, $p < .001$).

**Antisaccade latency.** The assumption of sphericity for the within-subjects factor was met based on the result of the Mauchly's sphericity test.

The 2 × 3 RMANOVA showed that neither Condition ($F$ (1, 36) = 2.20, $p = .147$, $\eta_p^2 = .058$) nor Group ($F$ (2, 36) = 1.39, $p = .262$, $\eta_p^2 = .072$) had a main effect on antisaccade latency. Furthermore, there was not a significant Condition × Group interaction ($F$ (2, 36) = 0.71, $p = .500$, $\eta_p^2 = .038$) either (Fig 3A). Descriptive statistics are provided in Table 3.

**Antisaccade error rate.** The assumption of sphericity for the within-subjects factor was met based on the result of the Mauchly's sphericity test.

The 2 × 3 RMANOVA yielded significant main effects of Condition ($F$ (1, 36) = 8.19, $p = .007$, $\eta_p^2 = .185$) on antisaccade error rate. Post-hoc comparisons revealed that the error rate was higher in the noise distraction condition ($p = .007$). Group also had a main effect ($F$ (2, 36) = 3.998, $p = .027$, $\eta_p^2 = .181$). Post-hoc comparisons revealed that the error rate of the motivational self-talk group was significantly higher than the control group ($p = .028$) and instructional self-talk group ($p = .014$). Additionally, there was a significant Condition × Group interaction ($F$ (2, 36) = 3.81, $p = .032$, $\eta_p^2 = .175$). Bonferroni comparison tests showed that the motivational self-talk group had a higher error rate in the noise distraction condition than the quiet condition ($p = .031$), participants who used instructional self-talk had a higher error rate in the noise distraction condition compared to the quiet condition ($p = .003$), and the control group did not demonstrate a significant difference in error rate between the quiet and noise distraction conditions ($p = .601$) (see Fig 3B). Descriptive statistics are provided in Table 3.

**Pupil diameter.** The assumption of sphericity for the within-subjects factor was met based on the result of the Mauchly's sphericity test.

The 2 × 3 RMANOVA revealed a significant main effect of Condition on pupil diameter ($F$ (1, 36) = 78.21, $p < .001$, $\eta_p^2 = .685$). Post-hoc comparisons revealed that the pupil diameter was larger in the noise distraction condition ($p < .001$). However, there was not a significant main effect of Group ($F$ (2, 36) = 2.23, $p = .110$, $\eta_p^2 = .110$). In addition, there was not a significant Condition × Group interaction ($F$ (2, 36) = 1.09, $p = .346$, $\eta_p^2 = .057$) either (Fig 3C). Descriptive statistics are provided in Table 3.

## Discussion

This study explored the effect of noise distraction on attentional control and the moderating effect of self-talk. Two experiments were conducted, the Stroop paradigm was used in Study 1 to demonstrate the detrimental effect of noise distraction on attentional control. Although Study 2 used the Antisaccade paradigm to further demonstrate that noise distraction can also impair AC in basketball players, failed to demonstrate all kinds of self-talk strategy facilitate AC. These results fully supported hypothesis1(i.e., noise distraction weakens AC level). It demonstrated how noise distraction, in line with attentional control theory, altered the equilibrium between goal-directed (top-down) and stimulus-directed (bottom-up) attention systems. Noise distraction made participants preferred to allocate attentional resources to external and irrelevant stimuli, which in turn affects attentional control, showing that the balance between goal-directed and stimulus-directed attention systems is key to affecting AC [41]. One

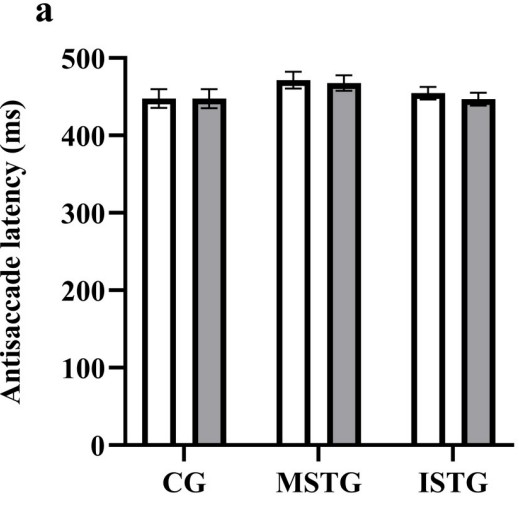

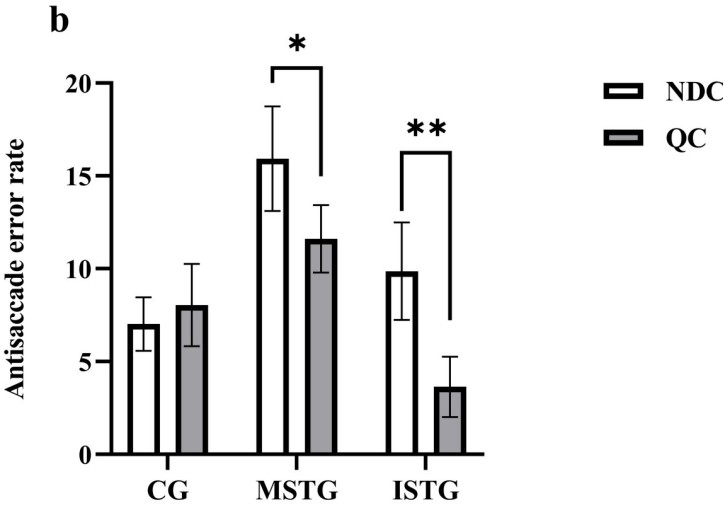

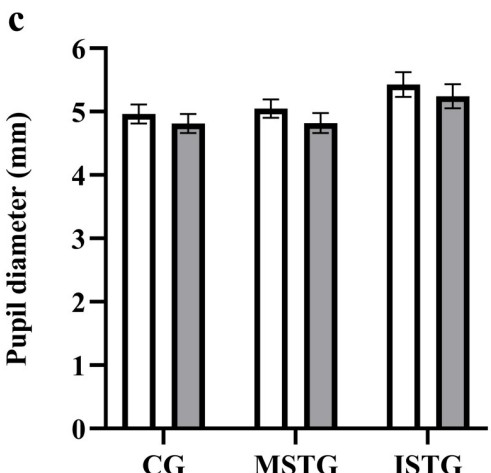

**Fig 3. Results of antisaccade task. a**, analysis results of antisaccade latency. **b**, analysis results of antisaccade error rate. **c**, analysis results of pupil diameter. Error bars represent standard error of mean (SEM). $*p < 0.05$, $**p < 0.01$.

indicator of cognitive load is pupil diameter, which was significantly higher in the noise distraction condition than in the quiet condition. This finding suggests that noise distraction not only increases bottom-up processing, but also cognitive load, which in turn impairs AC [8].

**Table 3. Result of the antisaccade task.**

|  | CG (*n* = 13) | | MSTG (*n* = 13) | | ISTG (*n* = 13) | |
|---|---|---|---|---|---|---|
|  | QC *M* (*SD*) | NDC *M* (*SD*) | QC *M* (*SD*) | NDC *M* (*SD*) | QC *M* (*SD*) | NDC *M* (*SD*) |
| Antisaccade latency(ms) | 447.52 (43.91) | 447.71 (43.01) | 467.83 (35.77) | 471.43 (38.79) | 446.98 (29.69) | 454.68 (29.60) |
| Antisaccade error rate | 8.04 (8.01) | 7.02 (5.20) | 11.6 (6.55) | 15.93 (10.15) | 3.64 (5.86) | 9.87 (9.49) |
| Pupil diameter(mm) | 4.81 (0.54) | 4.96 (0.54) | 4.82 (0.57) | 5.05 (0.53) | 5.24 (0.68) | 5.43 (0.70) |

CG: control group; MSTG: motivational self-talk group; ISTG: instructional self-talk group; NDC: noise distraction condition; QC: quiet condition.

However, it can be concluded that hypothesis 2 was only partially supported, as evidenced by the observed differences in antisaccade performance between the motivational self-talk group and the instructional self-talk and control groups. Specifically, the antisaccade error rate of the motivational self-talk group was worse than that of the other two groups. This seems to contradict previous studies on the benefits of self-talk for cognitive function [24]. Participants who underwent a self-talk intervention performed better in the old "pong" game (i.e., a task reflected attention ability) compared to control group under distracting condition [22]. In addition, Gregersen et al. (2017) demonstrated that contrary to the control group, participants who used self-talk in ego depletion state performed better on the visual test in terms of the proportion of correctly responses, and they demonstrated shorter reaction times on both the visual and the auditory tests [35]. Galanis et al. (2021) further investigated the relation between self-talk and attention via the Vienna Test System, the results suggest that self-talk benefits the attention functions and support postulations for an attentional interpretation of the facilitating effects of self-talk strategies on task performance [42]. However, the above studies failed to distinguish between motivational self-talk and instructional self-talk, and referred to them collectively as "self-talk". This may make it difficult to directly compare their findings to the present research, and it would be challenging to determine which type of self-talk was responsible for the observed effect. Therefore, the present research may contribute to a better understanding of the distinct effects of motivational and instructional self-talk on AC.

Ariely et al. (2009) conducted three experiments and found that very high incentives could led to a decrement in performance than moderate incentives [43]. Mobbs et al. (2009) further explained the relationship between motivation and performance through excessive drive and arousal (i.e., overmotivation theories) [44]. In this study, they presented incentive-dependent performance decline in a reward-pursuit task and participants had lower success rates in capturing a high valuable reward in a computer maze [44]. It indicated that high motivation from reward can impair performance even in a simple motor task and can also lead to a narrow focus on achieving the desired outcome at the expense of other important factors, such as skill development and enjoyment of the activity. Neuroimage results revealed that more activation of ventral midbrain which associated with incentive motivation and reward responding when the number of near-misses associated with high reward is coming [44]. Analysis in conjunction with pupil diameter revealed that using motivational self-talk did not result in higher cognitive load for participants. Therefore, the higher level of achievement motivation among the athlete group and the increased motivation level via motivational self-talk may be the reason why participants who used motivational self-talk performed worse in AC [45]. Moreover, the use of motivational self-talk was accompanied by high error rates in both quiet and noise distraction conditions, and this negative effect was more pronounced in the high cognitive load caused by the noise distraction. This may be due to the fact that high motivation is often accompanied by a specific action tendency, such as specialising in the pursuit of a particular goal, which in turn reduces attentional breadth [46, 47]. Breadth of attention is an important factor in the Antisaccade task because it reflects an individual's ability to maintain attentional focus on the central fixation point while also monitoring the periphery of their visual field for the appearance of the target stimulus. This requires individuals to have a broad attentional focus and the ability to suppress distractions that may interfere with their task performance. In addition, high motivation, whether it is approaching motivation or avoiding motivation, can affect cognitive flexibility. For instance, even with controlled arousal, high motivation has been shown to increase reaction time in shifting tasks [48]. Cognitive flexibility is also an important factor for AC, with individuals who possess higher cognitive flexibility tending to perform better on AC [49]. At last, athletes generally have a higher level of motivation than the general population, which is one of their special characteristics [50, 51]. In conclusion, one possible

reason why basketball players who used motivational self-talk performed worse in AC than other groups could be the combination of their pre-existing high motivation as athletes and the added high motivation from motivational self-talk, which may have resulted in a decrease in AC.

Motivational self-talk and effort instruction (i.e., an instruction that prompts participants to try harder in the current trial) share many similarities and have the common goal of improving motivation and performance. Specifically, effort instruction resulted in a global speed-up of responses, while the error rate is higher than stander condition in short foreperiod (FP), and motivational self-talk also increased the error rate in Antisaccade task [52]. Notably, both the short FP and the interval between "ready" (the signal of starting self-talk) and fixation point (the signal of starting the Antisaccade task) in the Antisaccade task are 1000ms. Therefore, we assume that the results of present study partially confirm that motivational self-talk in short FP hampered the precise control of the motor-system components that are decisive for performance [52–54].

The results from the descriptive statistics showed that individuals had less antisaccade errors than motivational self-talk group and control group when using instructional self-talk in quiet condition (although failing to reach a statistically significant difference). This result is consisted with previous studies and suggests that to some extent instructional self-talk has a facilitating effect on AC. It is possible that instructional self-talk is more effective for fine tasks than motivational self-talk, while motivational self-talk is more effective for gross tasks, which may mean that in terms of cognitive tasks, motivational self-talk is directed more towards perseverative self-control tasks, whereas instructional self-talk is directed more towards inhibitory self-control tasks [24]. Another possibility may be that different self-talk cues led participants to use different strategies when performing the task. According to the Strategic Resource Model, strategic allocation of cognitive resources can be modulated by the type of self-talk [55]. With instructional self-talk facilitating a more efficient distribution of attentional capacity compare to motivational self-talk. These findings underscore the significance of tailoring self-talk strategies in alignment with the strategic-resource framework to optimize cognitive performance in specific tasks [56].

Cognitive load is also an important factor that affects attentional control [9, 10, 57, 58]. The results of Study 2 demonstrated that noise distraction significantly increased cognitive load and reduced the saccade error rate. However, it is worth noting that, based on descriptive statistics, individuals had the highest cognitive load when using instructional self-talk. Nonetheless, under quiet conditions, the saccade error rate of the instructional self-talk group was significantly lower than that of the motivational self-talk group. Therefore, it is particularly important to consider the valence of cognitive load, that is, whether the increase in cognitive load is related to threatening stimuli. If the increase in cognitive load is brought by task-related stimuli, it may not affect the balance between attentional control systems. However, if it is brought by internal or external threatening stimuli, it may disrupt the balance between goal-directed and stimulus-directed attentional systems, thereby affecting AC.

In conclusion, a model integrating noise distraction, self-talk and AC is proposed. Noise distraction increases cognitive load and influence attentional control, with self-talk playing a modulatory role. Improper use of self-talk can disrupt the balance of attentional systems, diminish attentional control, and ultimately impact athletic performance, regardless of noise distraction (Fig 4).

This study is innovative in terms of ACT. First, by focusing on auditory stimuli from external sources, this study supports and advances the attentional control theory while also generating fresh ideas for future attentional control theory research. Second, the present study

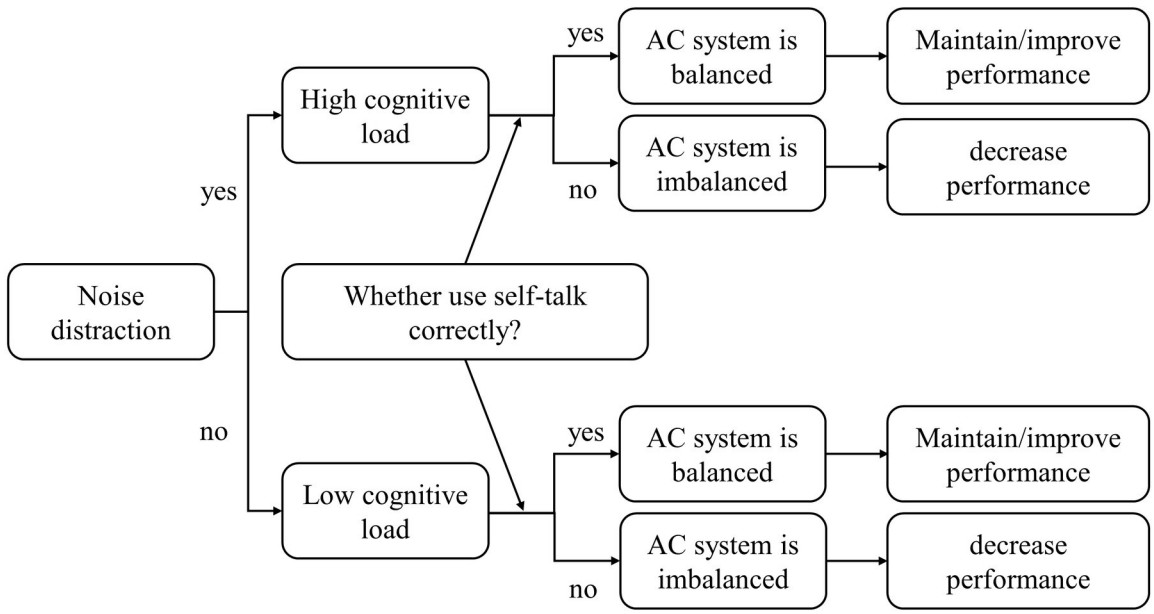

**Fig 4. ND-ST-AC integration model.** ND: noise distraction; ST: self-talk; AC: attentional control. Noise distraction increases cognitive load and affects AC, which is modulated by self-talk. Improper use of self-talk can disrupt AC system balance and thus affect task performance.

provides guidance on the use of self-talk as a psychological skill for basketball players to maintain AC under distraction conditions.

There are several limitations in this study. First, lack of measurement of control variables such as motivation and state anxiety. According to ACT, individuals with high anxiety often improve motivation as a compensation [41]. However, anxiety and motivation were not measured and controlled effectively in this study, and although participants did not report the presence of anxiety in the post-experimental interview, it does not mean that anxiety did not necessarily occur, and it is recommended that variables such as state anxiety, trait anxiety and motivation should be monitored in subsequent studies. Secondly, although the sample size met the requirements of G*power and the number of participants per group was higher than some of the self-talk intervention studies with athletes [22, 31, 59, 60], there is still a need to include a larger sample for empirical studies in the future to ensure stability and reproducibility of the results. Thirdly, due to an oversight in the experimental design and Stroop task programming, we included data only from the incongruent condition. As a result, the results of Study 1 do not capture the complete Stroop effect as traditionally calculated. Future studies would benefit from incorporating data from both congruent and incongruent conditions to offer a more comprehensive understanding of noise distraction's impact on the Stroop task. Finally, the participants in Study 2 were all male basketball players and there was a lack of female players, therefore caution is needed when extrapolating the findings to females. It is advised that female athletes be included in order to examine any potential gender-based differences.

## Supporting information

**S1 Data.**
(XLSX)

**S2 Data.**
(XLSX)

## Author Contributions

**Conceptualization:** Liu Yang, Yingchun Wang.

**Data curation:** Liu Yang.

**Formal analysis:** Liu Yang.

**Investigation:** Liu Yang.

**Methodology:** Liu Yang.

**Project administration:** Yingchun Wang.

**Software:** Liu Yang.

**Supervision:** Yingchun Wang.

**Validation:** Yingchun Wang.

**Visualization:** Yingchun Wang.

**Writing – original draft:** Liu Yang.

**Writing – review & editing:** Yingchun Wang.

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
