## [Decision Letter · Decision Letter 0]

24 Apr 2023

PONE-D-22-31246The effect of motivational and instructional self-talk on attentional control under noise distractionPLOS ONE

Dear Dr. Yang,

Thank you for submitting your manuscript to PLOS ONE. After careful consideration, we feel that it has merit but does not fully meet PLOS ONE’s publication criteria as it currently stands. Therefore, we invite you to submit a revised version of the manuscript that addresses the points raised during the review process.

We look forward to receiving your revised manuscript.

Kind regards,

Ruth Sarah Ogden, PhD

Academic Editor

PLOS ONE

Reviewers' comments:

Reviewer's Responses to Questions

**Comments to the Author**

1. Is the manuscript technically sound, and do the data support the conclusions?

Reviewer #1: Yes

Reviewer #2: Yes

2. Has the statistical analysis been performed appropriately and rigorously? 

Reviewer #1: Yes

Reviewer #2: Yes

3. Have the authors made all data underlying the findings in their manuscript fully available?

Reviewer #1: Yes

Reviewer #2: Yes

4. Is the manuscript presented in an intelligible fashion and written in standard English?

Reviewer #1: Yes

Reviewer #2: Yes

5. Review Comments to the Author

Reviewer #1: Comments to the Author:

General comments:

This study explored the effects of noise distraction on attentional control in basketball players and the moderating effects of self-talk. I think this study has practical implications. Firstly, the attentional control theory was verified in a specific sport (basketball). Secondly, the results were validated using eye-movement equipment. Finally, no positive results were found is insightful, but the authors do not discuss it in depth.

Specific comments:

Authors need to enhance the writing norms:

1. Line numbers should be added throughout.

2. Space before and after the symbols ±, =, etc.

3. English writing should be left-aligned, not end-to-end.

4. Pay attention to the tenses.

5. Note the subject of a sentence.

6. Please distinguish between full and half corner punctuation, e.g., for the Pupil diameter section of P16, please use a half corner comma.

About abstract:

The abstract should include purpose, methods, results, and conclusions.

1. There is too much "purpose". Consider stating that basketball players' attentional control affects performance, especially in the presence of noise distractions. Improving attentional control is therefore necessary. Self-talk is a psychological skill that enhances motor and cognitive performance and may improve attentional control. Thus, our study investigated the effects of noise distractions on attentional control and the moderating effect of self-talk.

2. Missing "Method", consider adding: (a) How does noise distractions operate? (b) What did the Stroop task and the Antisaccade task measure respectively? (c) Were instrumental and motivational self-talk used in each experiment?

3. The "conclusions" are not sufficiently illuminating. The current conclusions merely repeat the results. And your study did not find these findings.

About introduction:

1. The first paragraph is too long. Consider stopping after the introduction of attentional control theory and discussing the effect of noise on attentional control in the next paragraph.

2. One of the innovative points of this study is to examine the effect of noise distractions on attentional control and the moderating effect of self-talk among basketball players. Therefore, there is a need to emphasise the specificities of basketball, such as where does basketball need to focus attention? What distractions need to be inhibited more than anything else?

3. d values in italics, spaces before and after the “ = ”; what is the exact value of the 95% CI for the meta-analysis of Szalma et al. (2011)?

4. The paragraph introducing the quiet eye is oddly placed, splitting the review of the effects of noise distractions on attentional control. It could be considered for placement after the literature (Derakshan et al., 2009).

5. Note that the key words noise interference, noise disturbance, noise distraction should not be substituted at random.

6. Regarding the mechanisms by which self-talk affects motor performance, at citation points [22-26], it should be emphasized or summarized that the mechanisms are improvements in attention rather than listed theories and research findings.

7. "Hase et al. (2019) found motivated self-talk to be better due to participants' greater willingness to perceive threat as a challenge." What is the illumination of this finding for your research? One might consider adding: the study suggests that whether participants perceive the task as a threat or a challenge influences the performance of self-talk types on fine motor control.

8. The mechanism by which self-talk improves basketball performance needs to be highlighted. And it needs to be stressed that attentional control is key.

9. Please explain why two paradigms are used for study 1 and study 2?

10. The introduction needs to explain what can be examined in study 1 and study 2 respectively.

11. Is “Participant” a normal university student?

12. Hypotheses 2 and 3 should count as one hypothesis: self-talk moderates the effect of noise distractions on attentional control. Specifically, with noise distractions, only instrumental self-talk group improved attentional control compared to the other groups.

13. The use of different participants and different paradigms made the logic of the two experiments incoherent. And as previous studies have demonstrated that noise can affect attentional control, study 1 alone can be considered as a basis for testing the validity of the material.

Regarding study 1:

1. Are the participants normal university students? Need to specify.

2. Lowercase z in Hz.

3. Clarification is needed on how the results of the Stroop task can be interpreted. For example, in the inconsistent condition, shorter response times indicate stronger inhibition and smaller error rates indicate stronger inhibition.

4. Is the validity of noise material tested?

5. Instead of “greater”, use “more” to describe the size of a number in Data analysis.

6. Why use data within 2 standard deviations instead of 3?

7. For images and tables please use the insert form [insert figure 1], then the images and tables at the end of the manuscript.

8. Please use the 3-wire table

Regarding study 2:

1. Why did study 1 and study 2 use two different paradigms?

2. Is the qualified rate equal to the correct rate?

3. Were the sample sizes of study 1 and study 2 tested a priori power by G*power?

4. In Procedure (Page 13), it should be “study 1” and not “experiment 1”.

5. Stress assessment is a variable that the researcher should control for.

6. “The rest of group” can be replaced with “other groups” in the P15 manipulation check.

7. Note the writing specification, (F(1, 36) = 2.20, p =.147, ηp2 = .058), these data can be enclosed in parentheses.

8. Antisaccade error rate section, the reader is more interested in knowing the difference between motivational and instrumental self-talk in the presence of noise distractions. Rather than just reporting significant results, non-significant results may sometimes be more meaningful.

9. Has consideration been given to setting some additional variables as covariates to control for the interference? For example, the Stressor Appraisal Scale (Schneider, 2008) was used to measure whether participants perceived the task as stressful or challenging.

10. Please italicise statistics, e.g., M, SD.

11. There are no table notes in Table 2. The abbreviation of each table must be indicated so that its meaning is clear when it appears independently.

12. Please standardise the font for captions and notes on tables and images.

13. The relationship between quiet eye and eye movement data needs to be explained.

About discussion:

1. “However only to a limited extent were hypotheses 2 and 3 supported.” is not quite the appropriate sentence, because it is not supported.

2. “Specifically, while there was a significant main effect of self-talk on antisaccade error rate, post-hoc comparisons revealed that antisaccade error rate of motivational self-talk group was significantly higher than control group (p =.028) and instructional self-talk group (p =.014).” should be in the results section, not the discussion section.

3. These studies “Galanis et al. (2018), Gregersen et al. (2017), Galanis et al. (2021)” can only show that self-talk improves attentional control. You need to discern why your study did not find that self-talk improves attentional control. You need to clarify the differences between previous study and your study, which may be that the task is different in difficulty, or that certain confounding variables have not been controlled for.

4. The focus needs to be on “self-talk as an internal disturbance”.

5. Is high motivation equal to motivational self-talk? I think high motivation is the outcome and motivational self-talk is the method. I do not think that the literature that exemplifies how high motivation impairs attentional control proves that motivated self-talk is bad. If you think my understanding is incorrect, please illustrate with examples of literature.

6. The strength of your study is the addition of eye-movement data to prove your research hypothesis, but this aspect of the explanation is rarely mentioned in the discussion and needs to be added.

7. “Second, the present study provides guidance on the use of self-talk as a psychological skill for basketball players to improve AC”. Your study did not prove this, but only found that self-talk impairs attentional control. So, you should stress that self-talk can be a distraction and in practice you need to be aware that not all mental skills are applicable.

8. I see that your limitations mentioned not satisfying G*power, but in the methods section also mention how much sample size should be used for the calculation using G*power. In addition, you can perform a post hoc power calculations to determine the power.

Reviewer #2: p.5. The authors are advised to add in the self-talk literature that there is empirical evidence that self-talk strategies are effective in enhancing sport/task performance through systematic (Tod et al., 2011) and meta-analytic (Hatzigeorgiadis et al., 2011) reviews. Also regarding the matching hypotheses the authors are advised to consider two more matching hypotheses; one involving the setting by self-talk type matching and one involving the learning stage by self-talk type matching. Regarding the former, motivational self-talk seems more appropriate in competitive settings, whereas instructional self-talk seems more appropriate in training settings(Hatzigeorgiadis, Galanis, et al., 2014). Regarding the latter, instructional self-talk should be more effective for novel tasks, or tasks at the early stages of learning, whereas motivational self-talk should be more effective for well-learned tasks, or tasks at the automatic stage of performance (Zourbanos, Hatzigeorgiadis, Bardas, & Theodorakis, 2013).

p.7. Although in the introduction the authors write also about motivational self-talk in the hypotheses motivational self-talk is not clearly reported.

p.8 Why the authors didn’t use control group in the experimental design. Can this added in the limitation section?

p.10. Are the participants in experiment 2 participated also in experiment 1?

p.11. Have the authors tested the assumptions of the mixed design?

p.13. Why did you use these self-talk cues? How did you select them? The authors are advised to give more details about the self-talk cues selection.

p.13. Regarding the manipulation check, did you ask the participants if they said something else except of the instructional or motivational self-talk cues?

p. 14. Have the authors tested the assumptions of 2x3 repeated measures anova?

p.15. Please report the results of the assumptions of 2x3 Manova?

p.16. I suggest the authors to report for all the analyses performed the appropriate assumptions.

p.22. The authors are advised to include applied implications to their discussion section as this would help coaches but also sport psychologists to choose the right self-talk cues for their athletes.

6. PLOS authors have the option to publish the peer review history of their article (what does this mean?). If published, this will include your full peer review and any attached files.

Reviewer #1: **Yes: **Yu-Bu Wang

Reviewer #2: No

---

## [Author Response · Author response to Decision Letter 0]

11 May 2023

Response to reviewers

Dear Editor and Reviewers,

We would like to thank the reviewers for carefully reading our manuscript. We appreciate the comments and suggestions. In the following, we include a point-by-point response to the comments from each reviewer. In the revised manuscript, all the changes have been highlighted in red.

Reviewer #1: Comments to the Author:

General comments:

This study explored the effects of noise distraction on attentional control in basketball players and the moderating effects of self-talk. I think this study has practical implications. Firstly, the attentional control theory was verified in a specific sport (basketball). Secondly, the results were validated using eye-movement equipment. Finally, no positive results were found is insightful, but the authors do not discuss it in depth.

Specific comments:

Authors need to enhance the writing norms:

1. Line numbers should be added throughout.

Thank you for your suggestion. However, I did submit a manuscript with the line numbers, maybe the version you reviewed doesn't have the line numbers. 

2. Space before and after the symbols ±, =, etc.

Thank you for your suggestion. I have added the space around all symbols.

3. English writing should be left-aligned, not end-to-end.

Thank you for your suggestion. I have changed the text into left-aligned.

4. Pay attention to the tenses.

Thank you for your suggestion. I have reviewed the tenses paragraph by paragraph and marked the changes in red.

5. Note the subject of a sentence.

Thank you for your suggestion. I have changed the grammar errors and marked the changes in red.

6. Please distinguish between full and half corner punctuation, e.g., for the Pupil diameter section of P16, please use a half corner comma.

Thank you for your suggestion. I have changed the mis-usings and marked the changes in red.

About abstract:

The abstract should include purpose, methods, results, and conclusions.

1. There is too much "purpose". Consider stating that basketball players' attentional control affects performance, especially in the presence of noise distractions. Improving attentional control is therefore necessary. Self-talk is a psychological skill that enhances motor and cognitive performance and may improve attentional control. Thus, our study investigated the effects of noise distractions on attentional control and the moderating effect of self-talk.

Thank you for your suggestion. I have removed redundant parts to decrease the repetition of the "purpose" and added a sentence that illustrates how noise from competitive conditions can affect the attentional control (AC) and performance of basketball players. Please see the revised abstract for details.

2. Missing "Method", consider adding: (a) How does noise distractions operate? (b) What did the Stroop task and the Antisaccade task measure respectively? (c) Were instrumental and motivational self-talk used in each experiment?

Thank you for your suggestion. (a) In the abstract, I have mentioned that participants were required to wear earphones to receive noise distraction; (b) Both the Stroop and Antisaccade task are means of measuring AC, as I mentioned in the last paragraph of “Introduction”(p.8): “AC will be assessed using the Stroop task and Antisaccade task, two well-established paradigms for investigating AC.” Hence, I added “In Study 1, the effects of noise distraction on AC were initially demonstrated” in abstract; (c) the self-talk interventions only used in study2. I think “In Study 1, the effects of noise distraction on AC were initially demonstrated” this sentence can not only tell readers what the task was measuring, but also tell them that self-talk was not used in Study1.

3. The "conclusions" are not sufficiently illuminating. The current conclusions merely repeat the results. And your study did not find these findings.

Thank you for your suggestion. I have rewritten the conclusion according the results: “In conclusion, noise distraction increases cognitive load and reduces AC. For basketball players, the implementation of instructional self-talk in quiet condition proved to be more effective in enhancing AC. However, the utilization of motivational self-talk in noise distraction condition diminished AC level.”

About introduction:

1. The first paragraph is too long. Consider stopping after the introduction of attentional control theory and discussing the effect of noise on attentional control in the next paragraph.

Thank you for your suggestion. I have started a new paragraph to discuss effect of noise on AC.

2. One of the innovative points of this study is to examine the effect of noise distractions on attentional control and the moderating effect of self-talk among basketball players. Therefore, there is a need to emphasise the specificities of basketball, such as where does basketball need to focus attention? What distractions need to be inhibited more than anything else?

Thank you for your suggestion. I have added three major information in p.5, the first one is illustrating two dimensions in sport; the second discusses what needs to be focused on in basketball, and the third explains what distractions need to be avoided in basketball. Please refer to the revised introduction for more details. 

3. d values in italics, spaces before and after the “ = ”; what is the exact value of the 95% CI for the meta-analysis of Szalma et al. (2011)?

Thank you for your suggestion. I have changed “d” in italics and added space around “ = ”, plus, I have added the exact value of the 95% CI . Detail in “mean effect size across the 191 cognitive tasks was d = -0.34 (95%CI = -0.42 to -0.25) with the highest effect size for high loudness, short duration, d = -0.68, (95%CI = -0.96 to -0.40).” 

4. The paragraph introducing the quiet eye is oddly placed, splitting the review of the effects of noise distractions on attentional control. It could be considered for placement after the literature (Derakshan et al., 2009).

Thank you for your suggestion. I have moved the paragraph which introduces the relation between quiet eye and performance after the first paragraph. Surprisingly, it seems to have made the logical chain more fluent. Thank you for this suggestion again.

5. Note that the key words noise interference, noise disturbance, noise distraction should not be substituted at random.

Thank you for your suggestion. I have replaced all “noise interference” and “noise disturbance” with “noise distraction”.

6. Regarding the mechanisms by which self-talk affects motor performance, at citation points [22-26], it should be emphasized or summarized that the mechanisms are improvements in attention rather than listed theories and research findings.

Thank you for your suggestion. I have summarized the mechanisms by which self-talk affects motor performance through attention and added this sentence after introducing the research findings: “In summary, it can be concluded that self-talk affects performance by AC, and the correct use of self-talk facilitates the dominance of the goal-directed attention system, leading to AC on the task at hand.” 

7. "Hase et al. (2019) found motivated self-talk to be better due to participants' greater willingness to perceive threat as a challenge." What is the illumination of this finding for your research? One might consider adding: the study suggests that whether participants perceive the task as a threat or a challenge influences the performance of self-talk types on fine motor control.

Thank you for your suggestion. This sentence is helpful to better illustrate the illumination of this finding for my research, and I have added it (details in p.7). 

8. The mechanism by which self-talk improves basketball performance needs to be highlighted. And it needs to be stressed that attentional control is key.

Thank you for your suggestion. At the beginning of our discussion, I hope we could build a consensus is that “key” means the most important thing. As I responded in suggestion 6, I have added description that self-talk affects performance by attention. However, there are lots of aspects that self-talk affects performance. I acknowledge that AC is the key in the present research, but other factors (e.g., motivation, motor execution) are equally important to basketball performance. Therefore, I highlighted the mechanisms by which self-talk affects motor performance through attention, but stressing that AC is the “key” of performance may not be necessary. 

9. Please explain why two paradigms are used for study 1 and study 2?

Thank you for your question. The reason why used two paradigms separately is avoiding common method biases. If I used the same task to measure AC in two studies could potentially lead to common method biases. This may lead to similar or correlated measurement errors that are specific to the task. Using different tasks to measure AC in two studies can help reduce the impact of common method biases.

10. The introduction needs to explain what can be examined in study 1 and study 2 respectively.

Thank you for your suggestion. Study1 and Study2 both examined AC level and I mentioned it in the last paragraph of Introduction: “AC will be assessed using the Stroop task and Antisaccade task, two well-established paradigms for investigating AC [8, 31, 32]. Study 1 will measure the effect of noise distraction on AC via the Stroop task, and the Study 2 will further examine the effects of noise distraction on basketball players and the moderating effect of self-talk via the Antisaccade task.”

11. Is “Participant” a normal university student?

Thank you for your question. The “participants” in hypothesis refer to all participants in Study 1 and Study 2. The participants in Study 1 are normal university students, and participants in Study 2 are basketball players, and I hypothesized that their AC will be affect by noise distraction, so I call them “participants” collectively.

12. Hypotheses 2 and 3 should count as one hypothesis: self-talk moderates the effect of noise distractions on attentional control. Specifically, with noise distractions, only instrumental self-talk group improved attentional control compared to the other groups.

Thank you for your suggestion. Hypotheses 2 and 3 are indeed redundant expressions, I have changed the original hypothesis as your suggestion. Please see the last paragraph of introduction for more details.

13. The use of different participants and different paradigms made the logic of the two experiments incoherent. And as previous studies have demonstrated that noise can affect attentional control, study 1 alone can be considered as a basis for testing the validity of the material.

Thank you for your suggestion. As I responded in Suggestion 9 and 10, the Stroop and Antisaccade task are well established paradigms for testing AC, and the purpose of using different paradigms is avoiding the common method biases. Therefore, using different paradigms may not made the logic of the two experiments incoherent. However, you are totally correct about considering Study 1 as a basis for testing the validity of the material, and this is the most important reason why I designed it.

Regarding study 1:

1. Are the participants normal university students? Need to specify.

Thank you for your question. Yes, they are normal university students. I have specified this in the part of participants.

2. Lowercase z in Hz.

Thank you for your suggestion. I have changed the mis-usings.

3. Clarification is needed on how the results of the Stroop task can be interpreted. For example, in the inconsistent condition, shorter response times indicate stronger inhibition and smaller error rates indicate stronger inhibition.

Thank you for your suggestion. I have added the interpretation of indicators in p.10.

4. Is the validity of noise material tested?

Thank you for your question. As I responded in Suggestion 13, the purpose of Study 1 is testing the validity of noise material. The error rate was higher under noise distraction condition, it indicated that the material is effective.

5. Instead of “greater”, use “more” to describe the size of a number in Data analysis.

Thank you for your suggestion. I have replaced “greater” with “more”.

6. Why use data within 2 standard deviations instead of 3?

Thank you for your suggestion. There are two reasons why I consider 2 and -2 as thresholds. First, a z-score threshold of 2 is more commonly used than 3 generally, because using a z-score threshold of 3 would result in a more conservative criterion for identifying outliers, as it would require data points to be more extreme than with a z-score threshold of 2. Using data between 2 and -2 z-score can increase the reliability of statistical analyses by reducing the influence of extreme values on the results. Second, there may be situations where a z-score threshold of 3 is appropriate, such as when the data are highly skewed. However, the distribution of data is normal, so the use of 3 as threshold is not considered.

7. For images and tables please use the insert form [insert figure 1], then the images and tables at the end of the manuscript.

Thank you for your suggestion. However, I insert images and tables follow the “PLOS Manuscript Body Formatting Guidelines”.

8. Please use the 3-wire table

Thank you for your suggestion. However, the formal of tables follows “PLOS Manuscript Body Formatting Guidelines”.

Regarding study 2:

1. Why did study 1 and study 2 use two different paradigms?

Thank you for your question. I have answered this question in suggestion 9 and 13.

2. Is the qualified rate equal to the correct rate?

Thank you for your question. Yes, the qualified rate is equal to the correct rate.

3. Were the sample sizes of study 1 and study 2 tested a priori power by G*power?

Thank you for your question. The sample sizes were calculated before the experiments, and the relevant information has been added to the "participants" section. Specifically, 34 participants were included in Study 1 and 36 participants were included in Study 2, and all sample sizes were met.

4. In Procedure (Page 13), it should be “study 1” and not “experiment 1”.

Thank you for your question. I have changed this mistake and marked it as red.

5. Stress assessment is a variable that the researcher should control for.

Thank you for your feedback and suggestion. While we did not control for stress in this study, we acknowledge that stress can be an important factor to consider in present research. In hindsight, we agree that controlling for stress could have strengthened the internal validity of our study. Therefore, the first limitation we proposed in discussion is lacking measurements of control variables.

In future research, we plan to control for stress by implementing a stress induction procedure or by measuring participants' stress levels prior to the experiment. We will also report any stress-related variables in our study to better understand how stress may have influenced our results. Thank you for your valuable feedback and I will make sure to address this issue in future research.

6. “The rest of group” can be replaced with “other groups” in the P15 manipulation check.

Thank you for your suggestion. I have replaced “the rest of groups” with “other groups” in p.17.

7. Note the writing specification, (F(1, 36) = 2.20, p =.147, ηp2 = .058), these data can be enclosed in parentheses.

Thank you for your suggestion. I agree that enclosing these data in parentheses would improve the clarity and readability of the manuscript. I will make the necessary changes accordingly in the revised version of the manuscript.

8. Antisaccade error rate section, the reader is more interested in knowing the difference between motivational and instrumental self-talk in the presence of noise distractions. Rather than just reporting significant results, non-significant results may sometimes be more meaningful.

Thank you for your suggestion. I agree that this is an important point to clarify for the reader and I believe that it is still important to report this finding as it provides important information for future research in this area. I will revise this section to include this information. Please see more details in p.19.

9. Has consideration been given to setting some additional variables as covariates to control for the interference? For example, the Stressor Appraisal Scale (Schneider, 2008) was used to measure whether participants perceived the task as stressful or challenging.

Thank you for the suggestion to set additional variables as covariates to control for interference. However, due to limitations in our study design, I am unable to collect data on those variables. As a result, I am not able to include them as covariates in our analysis. I acknowledge that this may be a limitation of our study and will address it in future research.

10. Please italicise statistics, e.g., M, SD.

Thank you for your suggestion. I agree that italicizing statistics such as M and SD will enhance the clarity and readability of the manuscript, and I have made the necessary changes accordingly.

11. There are no table notes in Table 2. The abbreviation of each table must be indicated so that its meaning is clear when it appears independently.

Thank you for your comment regarding Table 2. I appreciate your feedback and recognize the importance of indicating the meaning of abbreviations used in the table. I have added table notes to clarify the meaning of abbreviations and improve the readability of the table. 

12. Please standardise the font for captions and notes on tables and images.

Thank you for your suggestion. I have revised our manuscript to ensure that all captions and notes have the same font size and type.

13. The relationship between quiet eye and eye movement data needs to be explained.

Thank you for your question. Although quiet eye and eye movements in the antisaccade task are both indicators measure AC, thy are two different concepts related to eye behavior.

Quiet eye refers to the final fixation of the eyes on a specific target before initiating a motor action, such as throwing a ball or shooting a target in sports. During the quiet eye phase, the athlete's gaze is fixed on a specific location, and this gaze fixation is associated with improved performance accuracy.

On the other hand, eye movements in the antisaccade task refer to a cognitive task that requires inhibiting a reflexive saccade toward a suddenly appearing visual stimulus and instead making a voluntary eye movement to the opposite side. The task requires inhibitory control and cognitive flexibility, and performance on this task is used as an index of executive function.

Therefore, the key difference between quiet eye and eye movements in the antisaccade task is that the former refers to a gaze fixation behavior associated with improved motor performance accuracy, while the latter refers to a cognitive task requiring inhibitory control and cognitive flexibility.

About discussion:

1. “However only to a limited extent were hypotheses 2 and 3 supported.” is not quite the appropriate sentence, because it is not supported.

Thank you for your suggestion. I have changed another expression based on new hypothesis 2 that according to Suggestion 12. Please see more details in p.20.

2. “Specifically, while there was a significant main effect of self-talk on antisaccade error rate, post-hoc comparisons revealed that antisaccade error rate of motivational self-talk group was significantly higher than control group (p =.028) and instructional self-talk group (p =.014).” should be in the results section, not the discussion section.

Thank you for your suggestion. I agree that the information regarding the significant main effect of self-talk on antisaccade error rate and the post-hoc comparisons should be included in the Results section rather than the Discussion section. I have revised the manuscript accordingly to reflect this change. Please see more details in p.18 and p.21.

3. These studies “Galanis et al. (2018), Gregersen et al. (2017), Galanis et al. (2021)” can only show that self-talk improves attentional control. You need to discern why your study did not find that self-talk improves attentional control. You need to clarify the differences between previous study and your study, which may be that the task is different in difficulty, or that certain confounding variables have not been controlled for.

Thank you for your valuable feedback. I agree that it is important to clarify the differences between our study and previous studies that have found self-talk to improve attentional control. I have added two sentences in p.22 to address this reason. While it is difficult to directly compare the task difficulties between studies, our study was unique in that we distinguished between motivational and instructional self-talk interventions. This may have contributed to the partially different findings compared to previous studies. Thank you again for your helpful comments.

4. The focus needs to be on “self-talk as an internal disturbance”.

Thank you for your suggestion. I apologize for misapplying the interpretation of self-talk as an internal disturbance in the present research. It is important to note that self-talk can be either positive or negative, and its impact on an individual's mental state and performance can depend on the content and context of the self-talk. While some types of negative self-talk, such as self-criticism or rumination, can contribute to feelings of distress or inner disturbance, the types of self-talk used in the present research were motivational and instructional self-talk, which are typically viewed as positive forms of self-talk.

Regarding the reference I quoted, Brinthaupt (2019) suggested that although self-talk frequency is positively correlated with cognitive disruption, the use of self-talk is generally viewed as a self-regulatory tool. Additionally, Ariel (2022) found that some individuals may view self-talk as an inner disturbance due to the social stigma associated with it as non-normative behavior in adulthood. However, it is important to recognize that self-talk is a natural and important cognitive process that individuals use to regulate their thoughts and emotions (Ariel, 2022).

Therefore, while self-talk can sometimes be viewed as an internal disturbance, it is not a correct interpretation in the present research. I have deleted this expression from the manuscript, and I appreciate your feedback and guidance in improving the clarity of our interpretation.

5. Is high motivation equal to motivational self-talk? I think high motivation is the outcome and motivational self-talk is the method. I do not think that the literature that exemplifies how high motivation impairs attentional control proves that motivated self-talk is bad. If you think my understanding is incorrect, please illustrate with examples of literature.

Thank you for your question. You are correct that high motivation is an outcome of motivational self-talk, as motivational self-talk tends to increase motivation. In the present research, basketball players who used motivational self-talk performed worse in the Antisaccade task compared to other groups. One possible explanation for this is that the high motivation brought about by self-talk may have influenced attentional control (AC).

First, athletes generally have a higher level of motivation than the general population, which is one of their special characteristics (Giakoni-Ramírez et al., 2022; Šmela et al., 2017). 

Second, according to overmotivation theory (also known as Incentive-based theory), people are motivated to engage in certain behaviors or activities because they believe that doing so will lead to a desirable outcome or reward. However, larger and more desirable rewards are more likely to motivate people than smaller or less desirable ones. Overmotivation from rewards can impair performance even in simple motor tasks, and can also lead to a narrow focus on achieving the desired outcome at the expense of other important factors, such as skill development and enjoyment of the activity (Mobbs et al., 2009; Short & Sorrentino, 1986).

Third, there is evidence to suggest that high approach-motivation reduces the breadth of attention (Gable & Harmon-Jones, 2008). Breadth of attention is an important factor in the Antisaccade task because it reflects an individual's ability to maintain attentional focus on the central fixation point while also monitoring the periphery of their visual field for the appearance of the target stimulus. This requires individuals to have a broad attentional focus and the ability to suppress distractions that may interfere with their task performance.

In conclusion, one possible reason that basketball players who used motivational self-talk performed worse in AC than other groups is the conjunction of high motivation in athletes and high motivation from self-talk interventions, which may have resulted in a decrease in AC.

Your question has helped me provide a better explanation in the discussion section regarding this phenomenon. Please see p.24 for more details.

6. The strength of your study is the addition of eye-movement data to prove your research hypothesis, but this aspect of the explanation is rarely mentioned in the discussion and needs to be added.

Thank you for your suggestion. The antisaccade task has some advantages compared to the Stroop task or traditional attentional control tasks. Firstly, the task requires a high degree of cognitive control, as it involves inhibiting a prepotent response (looking towards a peripheral target) and instead generating a saccade in the opposite direction. This makes it a more ecologically valid measure of attentional control than traditional tasks, as it more closely reflects the types of inhibitory processes required in real-world situations where we need to inhibit automatic responses. Second, the antisaccade task has been shown to be a reliable and valid measure of attentional control, with good test-retest reliability and sensitivity to individual differences in attentional control. the two core indicators of the antisaccade task are saccade latency and saccade error rate, which correspond to the reaction time and error rate of key presses in the Stroop task, other eye movement data (except pupil diameter) is meaningless to us. Pupil diameter is an additional indicator to measure cognitive load. 

However, I completely agree that discussing pupil diameter data is an essential aspect of our research, which adds strength to our study and proves our research hypothesis. I have added related discussion, please see p.24 to p.25 for more details

7. “Second, the present study provides guidance on the use of self-talk as a psychological skill for basketball players to improve AC”. Your study did not prove this, but only found that self-talk impairs attentional control. So, you should stress that self-talk can be a distraction and in practice you need to be aware that not all mental skills are applicable.

Thank you for your suggestion. I have changed the expression into “Second, the present study …… to maintain AC under distraction conditions.” And in the final of Discussion, I proposed ND-ST-AC Integration Model to better illustrate the results. Please see more details in p.25 and Fig 4.

8. I see that your limitations mentioned not satisfying G*power, but in the methods section also mention how much sample size should be used for the calculation using G*power. In addition, you can perform a post hoc power calculations to determine the power.

Thank you for your suggestion. I have included the information related to the sample size calculated by G*power. Furthermore, after performing a post hoc power analysis, Power (1-β err prob) = 0.768. This indicates that the current study has a relatively high probability of detecting a true effect or relationship, provided that such an effect or relationship exists in the population and the statistical test is appropriately designed and executed.

References

Ariel, N. (2022). Don’t think before you speak: on the gradual formation of thoughts during speech. Pedagogy, Culture & Society, 1-13. https://doi.org/10.1080/14681366.2022.2039270

Brinthaupt, T. M. (2019). Individual Differences in Self-Talk Frequency: Social Isolation and Cognitive Disruption. Frontiers in Psychology, 10(1088). https://doi.org/10.3389/fpsyg.2019.01088

Gable, P. A., & Harmon-Jones, E. (2008). Approach-Motivated Positive Affect Reduces Breadth of Attention. Psychological Science, 19(5), 476-482. https://doi.org/10.1111/j.1467-9280.2008.02112.x

Giakoni-Ramírez, F., Merellano-Navarro, E., & Duclos-Bastías, D. (2022). Professional Esports Players: Motivation and Physical Activity Levels. International Journal of Environmental Research and Public Health, 19(4), 2256. https://doi.org/10.3390/ijerph19042256

Mobbs, D., Hassabis, D., Seymour, B., Marchant, J. L., Weiskopf, N., Dolan, R. J., & Frith, C. D. (2009). Choking on the Money: Reward-Based Performance Decrements Are Associated With Midbrain Activity. Psychological Science, 20(8), 955-962. https://doi.org/10.1111/j.1467-9280.2009.02399.x

Short, J.-A. C., & Sorrentino, R. M. (1986). Achievement, affiliation, and group incentives: A test of the overmotivation hypothesis. Motivation and Emotion, 10(2), 115-131. https://doi.org/10.1007/BF00992251

Šmela, P., Pačesová, P., Kraček, S., & Hájovský, D. (2017). Performance Motivation of Elite Athletes, Recreational Athletes and Non-Athletes. Acta Facultatis Educationis Physicae Universitatis Comenianae, 57(2), 125-133. https://doi.org/10.1515/afepuc-2017-0012

Reviewer #2: p.5. The authors are advised to add in the self-talk literature that there is empirical evidence that self-talk strategies are effective in enhancing sport/task performance through systematic (Tod et al., 2011) and meta-analytic (Hatzigeorgiadis et al., 2011) reviews. Also regarding the matching hypotheses the authors are advised to consider two more matching hypotheses; one involving the setting by self-talk type matching and one involving the learning stage by self-talk type matching. Regarding the former, motivational self-talk seems more appropriate in competitive settings, whereas instructional self-talk seems more appropriate in training settings(Hatzigeorgiadis, Galanis, et al., 2014). Regarding the latter, instructional self-talk should be more effective for novel tasks, or tasks at the early stages of learning, whereas motivational self-talk should be more effective for well-learned tasks, or tasks at the automatic stage of performance (Zourbanos, Hatzigeorgiadis, Bardas, & Theodorakis, 2013).

Thank you for your suggestion. We appreciate your input and have revised our paper to include the empirical evidence of the effectiveness of self-talk strategies in enhancing sport/task performance through systematic and meta-analytic reviews (Tod et al., 2011; Hatzigeorgiadis et al., 2011).

While we agree with the two hypotheses you proposed, we would like to clarify that they may not be suitable for the present study due to two specific reasons. First, the condition settings in the present study are quiet and noise distraction conditions, which may not fully match with training and competitive settings. There may be external distractions in the training setting, while the quiet laboratory environment in our study meant that external distractions were well-controlled.

Second, in Study 2, during the practice stage, if the error rate of the first saccade was higher than 20%, a new 20-trial was required until the error rate was less than 20%. This means that all participants in Study 2 had well-learned the purpose of the antisaccade task, and therefore the hypotheses involving the learning stage may not match the present study design.

However, these two hypotheses you proposed were perfect to add to the introduction to better introduce motivational and instructional self-talk, thank you again for your insightful suggestion. Please see more details in p.6 to p.7. 

p.7. Although in the introduction the authors write also about motivational self-talk in the hypotheses motivational self-talk is not clearly reported.

Thank you for your suggestion. We have revised the manuscript to better illustrate the hypotheses at the end of Introduction, please see more details in p.9.

p.8 Why the authors didn’t use control group in the experimental design. Can this added in the limitation section?

Thank you for your suggestion. We appreciate your attention to detail and concern regarding our experimental design. We would like to clarify that we did, in fact, use a control group in our study. Participants in the control group performed the same task as the experimental group but did not receive any self-talk intervention. We apologize for any confusion or lack of clarity in our paper that may have led to this misunderstanding. We will make sure to clarify this in the revised manuscript. Thank you for bringing this to our attention.

p.10. Are the participants in experiment 2 participated also in experiment 1?

Thank you for your question. The participants in Experiment 2 did not participate in Experiment 1, as we recruited a new set of participants for each experiment. The participants in Experiment 1 were normal university students, the participants in Experiment 2 were national second-level basketball players. I have clarified this in the revised manuscript

p.11. Have the authors tested the assumptions of the mixed design?

Thank you for your question. A Mauchly's sphericity test was conducted to examine the assumption of sphericity for the within-subjects factor in a mixed design. The result showed that for all variables, the Mauchly's W value was 1.000, indicating that the assumption of sphericity was met. The approximate chi-square value was .000 with 0 degrees of freedom, but the significance level could not be calculated. Therefore, no adjustment was necessary and the traditional F-test could be used to analyze the data.

In conclusion, the assumption of sphericity for the within-subjects factor was met based on the result of the Mauchly's sphericity test.

p.13. Why did you use these self-talk cues? How did you select them? The authors are advised to give more details about the self-talk cues selection.

Thank you for your question. Thank you for your suggestion. We selected the self-talk cues based on the working definition of self-talk (Hardy, 2006) and previous literature (Abdoli et al., 2018; Dali & A. Parnabas, 2018). According to one of definitions from Hardy (2006), self-talk should serve at least two functions: instructional and motivational, for the athlete. Therefore, instructional self-talk cues and motivational self-talk cues were used in the present research.

“I can do it” or “I can” is a commonly used motivational self-talk cue that has been shown to enhance performance among basketball players (Abdoli et al., 2018; Dali & A. Parnabas, 2018). 

There are no references about the instructional self-talk cue on the antisaccade task. However, according to Hardy (2006), the performance of relatively fine motor tasks requiring skill, timing, and accuracy was enhanced to a greater extent by instructional self-talk, focusing on the technical aspects of performance, than motivational self-talk. Therefore, “Look at the opposite” is intended to remind participants to look in the opposite direction of the target, which is the goal and technical aspect of the task. 

We have added these details to the revised manuscript to provide more clarity on the selection of the self-talk cues.

p.13. Regarding the manipulation check, did you ask the participants if they said 

Thank you for your question. We did ask participants after the experiment whether they used any self-talk cues other than the ones provided to them (i.e., "I can do it" for the motivational self-talk group and "look at the opposite" for the instructional self-talk group). None of the participants reported using any other self-talk cues during the task. We have added this information to the manuscript in p.17.

p. 14. Have the authors tested the assumptions of 2x3 repeated measures anova?

Thank you for your question. Like I responded in “p.11. Have the authors tested the assumptions of the mixed design?”, the assumption of sphericity for the within-subjects factor was met based on the result of the Mauchly's sphericity test.

p.15. Please report the results of the assumptions of 2x3 Manova?

Thank you for your suggestion. Here is the report the results of the assumptions of 2x3 MANOVA. 

Antisaccade Latency: The 2 × 3 RMANOVA showed that neither Condition (F (1, 36) = 0.206, p = .651, η2 p = .069) nor Group (F (2, 36) = 2.651, p = .077, η2 p = .073) had a main effect on antisaccade latency. Furthermore, there was not a significant Condition × Group interaction (F (2, 36) = 0.066, p = .936, η2 p = .002) either

Antisaccade Error Rate: The 2 × 3 MANOVA yielded no significant main effects of Condition (F(1, 36) = 2.855, p = .095, η2 p = .038), but Group has a significant main effects (F(2, 36) = 6.309, p = .003, η2 p = .149) on antisaccade error rate. Post-hoc comparisons revealed that participants in the motivational self-talk group had a significantly higher error rate than those in the control group (p = .005) and instructional self-talk group (p = .002). There was no significant Condition × Group interaction (F(2, 36) = 1.329, p = .271, η2 p = .036). 

Key-Press Error Rate: The results of the 2 × 3 MANOVA on key-press error rate revealed no significant main effect of condition (F(1, 36) = 0.406, p = .526, η2 p = .006) or group (F(2, 36) = 0.216, p = .806, η2 p = .006). There was also no significant Condition × Group interaction (F(2, 36) = 0.183, p = .833, η2 p = .005).

Pupil Diameter: The 2 × 2 MANOVA on pupil diameter showed that no significant main effects of Condition (F(1, 36) = 1.932, p = .169, η2 p = .026), but Group has a significant main effects (F(2, 36) = 4.404, p = .016, η2 p = .109) on antisaccade pupil diameter. There was no significant Condition × Group interaction (F(2, 36) = 0.027, p = .973, η2 p = .001).

We also conducted a post hoc power analysis, Input: Effect size f(V) = 0.25, α err prob = 0.05, Total sample size = 39, Number of groups = 3, Number of measurements = 2. After calculating, Power (1-β err prob) = 0.249, this means that there is a low probability (24.9%) of correctly rejecting the null hypothesis when it is false. Therefore, MANOVA may not be an appropriate statistical method for the present research.

p.16. I suggest the authors to report for all the analyses performed the appropriate assumptions.

Thank you for your suggestion. We appreciate your feedback and will make sure to report the appropriate assumptions for all the statistical analyses performed in the revised manuscript. Please see more details in the section of Result of the Study 2.

p.22. The authors are advised to include applied implications to their discussion section as this would help coaches but also sport psychologists to choose the right self-talk cues for their athletes.

Thank you for your suggestion. In the final of Discussion, I proposed ND-ST-AC Integration Model (i.e., Noise Distraction-Self-Talk-Attentional Control Integration Model), the model integrates several factors, including noise distraction, self-talk, and attentional control, and their potential effects on cognitive load and athletic performance. The conclusion emphasizes the importance of using self-talk appropriately to balance attentional systems and maintain attentional control, particularly in the presence of noise distraction. Improper use of self-talk can disrupt the balance of attentional systems, diminish attentional control, and ultimately impact athletic performance, regardless of noise distraction.

References

Abdoli, B., Hardy, J., Riyahi, J. F., & Farsi, A. (2018). A Closer Look at How Self-Talk Influences Skilled Basketball Performance. The Sport Psychologist, 32(1), 9-15. https://doi.org/10.1123/tsp.2016-0162

Dali, M. S., & A. Parnabas, V. (2018). The effects of self-talk on free throw performance and the level of anxiety among male novice basketball players. MoHE, 7(1). https://doi.org/10.15282/mohe.v7i1.174

Hardy, J. (2006). Speaking clearly: A critical review of the self-talk literature. Psychology of Sport and Exercise, 7(1), 81-97.

---

## [Decision Letter · Decision Letter 1]

28 Jun 2023

PONE-D-22-31246R1The effect of motivational and instructional self-talk on attentional control under noise distractionPLOS ONE

Dear Dr. Wang,

Thank you for submitting your manuscript to PLOS ONE. After careful consideration, we feel that it has merit but does not fully meet PLOS ONE’s publication criteria as it currently stands. Therefore, we invite you to submit a revised version of the manuscript that addresses the points raised during the review process.

As can been seen from the reviews, one reviewer would like to see a small number of minor changes before acceptance. 

We look forward to receiving your revised manuscript.

Kind regards,

Ruth Sarah Ogden, PhD

Academic Editor

PLOS ONE

Journal Requirements:

Reviewers' comments:

Reviewer's Responses to Questions

**Comments to the Author**

1. If the authors have adequately addressed your comments raised in a previous round of review and you feel that this manuscript is now acceptable for publication, you may indicate that here to bypass the “Comments to the Author” section, enter your conflict of interest statement in the “Confidential to Editor” section, and submit your "Accept" recommendation.

Reviewer #1: All comments have been addressed

Reviewer #2: All comments have been addressed

2. Is the manuscript technically sound, and do the data support the conclusions?

Reviewer #1: Yes

Reviewer #2: Yes

3. Has the statistical analysis been performed appropriately and rigorously? 

Reviewer #1: Yes

Reviewer #2: Yes

4. Have the authors made all data underlying the findings in their manuscript fully available?

Reviewer #1: Yes

Reviewer #2: Yes

5. Is the manuscript presented in an intelligible fashion and written in standard English?

Reviewer #1: Yes

Reviewer #2: Yes

6. Review Comments to the Author

Reviewer #1: I accept all the changes you have made. However, there are some changes that still need to be made.

1. I seriously recommend professional language editing. I gave some examples.

L58-62. Sentence is too long and difficult to be understood.

P79. Remember that “studies” or “investigations” do not find anything, but “researchers” or “scholars” or “investigators” do!

L158. Please distinguish between affection and emotion. I think it should be emotion here.

Please use participants instead of subjects.

L202. The word "normal" you are trying to express should mean untrained, please amend it.

L310-319. This sentence is so long that it makes me very confused.

2. I still think the abstract is too long, please refer to the author's guidelines for changes.

3. Parameters should be reported with two or three decimals only.

4. It is also good practice to present the interaction effects first, before moving onto the main effects.

5. There are still many missing spaces, e.g., L130.

Reviewer #2: The authors have addressed all the comments provided by all the reviewers. I have no further comments.

7. PLOS authors have the option to publish the peer review history of their article (what does this mean?). If published, this will include your full peer review and any attached files.

Reviewer #1: **Yes: **Yu-Bu Wang

Reviewer #2: No

---

## [Author Response · Author response to Decision Letter 1]

5 Jul 2023

1. I seriously recommend professional language editing. I gave some examples.

L58-62. Sentence is too long and difficult to be understood.

Thank you for your valuable feedback. I understand the sentence was initially complex and challenging to follow. I have now broken it down into several smaller sentences to clarify the meaning and improve readability. The revised sentences now read as follows: “Furthermore, anxiety escalates the allocation of attention to irrelevant stimuli, both internal and external, leading to bottom-up processing. This process affects key functions of AC (i.e., inhibition and shifting), especially inhibition, the primary indicator.” I hope this provides a clearer understanding of the concept I am presenting.

P79. Remember that “studies” or “investigations” do not find anything, but “researchers” or “scholars” or “investigators” do!

Thank you for your valuable feedback. I understand your point and agree that it is the researchers who make the discoveries, not the studies themselves. I have therefore revised the sentences and you could see them in revised manuscript with track changes. I hope this modification appropriately addresses your concern.

L158. Please distinguish between affection and emotion. I think it should be emotion here.

Thank you for your suggestion. In L158, I cited the article from Hardy et al. (2009). In the original text, under the heading of “Affective mechanisms,” Hardy stated: “Although there has been much debate in the sport and exercise psychology literature concerning the definition and conceptual distinction of the terms ‘affect’, ‘mood’ and ‘emotion’, we have used the phrase ‘affective mechanisms’, as an umbrella term overseeing this family of concepts.” And after further checking on APA Dictionary of Psychology, the definition of “affect” is: Any experience of feeling or emotion, ranging from suffering to elation, from the simplest to the most complex sensations of feeling, and from the most normal to the most pathological emotional reactions. Hence, I believe “affect” is the most appropriate term to use in this context, and not “emotion” or “affection”.

Please use participants instead of subjects.

Thank you for your suggestion. I have changed all “subjects” into “participants”, you could see them in revised manuscript with track changes.

L202. The word "normal" you are trying to express should mean untrained, please amend it.

Thank you for your suggestion. You are correct that the term 'normal' was not the most accurate descriptor in this context. I have now amended the sentence to specify that the participants had no prior basketball training, which is what I intended to convey with the term 'normal'. The revised sentence could be seen in revised manuscript with track changes.

L310-319. This sentence is so long that it makes me very confused.

Thank you for your suggestion. I have revised the sentence to improve its clarity and readability. I believe this revision provides a clearer, more understandable description of the trial process.

2. I still think the abstract is too long, please refer to the author's guidelines for changes.

Thank you for your suggestion. I have revised and shortened the abstract, ensuring it aligns with the guidelines provided for authors. I hope that the changes made are satisfactory and meet your requirements. 

3. Parameters should be reported with two or three decimals only.

Thank you for your suggestion. I have kept 2 as decimals for all parameters in table two.

4. It is also good practice to present the interaction effects first, before moving onto the main effects.

Thank you for your suggestion. According to APA 7th edition, however, reporting statistical results needs to be relevant hypotheses. In addition, presenting the results of tests should be in the order that you performed them—report the outcomes of main tests before post-hoc tests. In the present research, the first hypothesis to test is whether the noise has an effect on the AC; the second hypothesis to test is whether the self-talk has an effect on the AC. The moderate effect is tested at the end. Therefore, in order to maintain the overall logic of the article, it is necessary to keep the original order of results presentation.

5. There are still many missing spaces, e.g., L130.

Thank you for your suggestion. I have thoroughly reviewed the document and corrected this issue not only at the line you pointed out, but throughout the entire manuscript. I have taken extra care to ensure the correct use of spaces throughout the text. I appreciate your patience and attention to detail, which helps improve the clarity and readability of the manuscript.

References 

Hardy, J., Oliver, E., & Tod, D. (2009). A framework for the study and application of self-talk within sport. Advances in applied sport psychology: A review, 37-74.

---

## [Decision Letter · Decision Letter 2]

24 Jul 2023

PONE-D-22-31246R2The effect of motivational and instructional self-talk on attentional control under noise distractionPLOS ONE

Dear Dr. Wang,

Thank you for submitting your manuscript to PLOS ONE. After careful consideration, we feel that it has merit but does not fully meet PLOS ONE’s publication criteria as it currently stands. Therefore, we invite you to submit a revised version of the manuscript that addresses the points raised during the review process.

 Editor comments. As the newly appointed editor responsible for handling this manuscript (the former editor is no longer available), I have read your manuscript and the comments from the two reviewers. To me, it seems the reviews generally converge towards a positive evaluation over the rounds, which aligns with my own impression of the current manuscript. However, there are several critical methodological points that urgently need to be clarified. These points mainly concern the calculation of effects and performance parameters in the tasks (Stroop task and Anti-saccade task) as well as the conceptual implementation of motivational self-talk. Overall, I have provided detailed comments that should be carefully addressed in a further revision. My comments are listed below.

We look forward to receiving your revised manuscript.

Kind regards,

Michael B. Steinborn, PhD

Section Editor

PLOS ONE

**Additional Editor Comments**

 (-1-) theory  The authors have utilised Eysenck's attention control theory of anxiety as the theoretical framework for the study. The main argument of this theory revolves around the notion that momentary (state) anxiety negatively impacts working memory functions, including executive control, phonological loop, and visual-spatial sketchpad. While this theory might be insightful and serve as a base metaphor in exploring individual differences in anxiety and related personality concepts such as neuroticism, it does not offer any specific predictions about the performance effects that could potentially be expected in the present research context. To address this limitation and enhance the theoretical impact of the study, it is recommended to provide a theoretical background that focuses on predicting the effects of the factors that are assumed to influence the outcome performance measures in the Stroop and Anti-saccade tasks. This theoretical background should address how the chosen factors, such as noise distraction and self-talk strategies, are expected to interact with attentional control and ultimately impact task performance. By incorporating a more predictive theoretical framework, the study findings and interpretations would be much better grounded and would likely contribute largely to the understanding of attentional control under noise distraction conditions. To guide you in the process of revision, I recommend a recent work of Schumann et al. which provides you with an overview of theory and relevant studies on the interplay of computational (directing, coordinating) and energetical (motivating) attentional resources to ensure performance stability (doi:10.3389/fpsyg.2022.867978), in both basic and applied (e.g., sports) psychology.   (-2-) design to study "self-talk" vs. design to study "cuing"   In sports psychology fields, the concept of self-talk is typically used to study real-life tasks such as tennis or basketball because in these tasks, the coordination of multiple mental operations is crucial as they need to be executed in a well-timed sequence and in the correct or optimal order. When the task is not already over-learned (or is below a certain level of automatisation), self-talk strategies are critical as they can aid coordinating action sequences. In other words, directional self-talk is equivalent to representing the points in a sequence in working memory prior to the actual execution of the action, ensuring that the intended sequence of movements is followed accurately. When the task is highly over-learned, then sole motivational self-talk is more essential (see Schumann et al., doi:10.3389/fpsyg.2022.867978, chap. 4.5) in ensuring the retrieval of specified sequences from memory (and not to miss it at some occasions). The present paper employs the concept and terminology of self-talk explicitly, while the actual experiments align more with other concepts referred to as "cueing" in the literature. The problem here is that this creates a misunderstanding as you link your experiments to the self-talk literature while the actual experiments are essentially designed as commonly done in the field of cognitive-experimental research. These studies focus on cueing effects, specifically involving directional cueing (e.g., look-away effect in the antisaccade task) and effort cueing (e.g., try-harder effect). To understand the mechanism of cueing, in particular to understand effort cuering relating to why some of the manipulations did not work in the present experimental context of your study, I would suggest one of my own work (doi:10.1007/s00426-016-0810-1), apologising for doing so, but in this work it is explained how try-harder manipulation work, what factors are critical and what aspects of performance is most indicative of increased motivation. In very brief, (1) motivational cues work best when pre-instructed (i.e., the cue try-hard is explained before the experiment to the participant), (2) when they are implemented as reminders during the task (because participants would forget the instruction over the course of a block of trials (see Schumann et al., 2022, chap. 5.4) when the instruction is only given before the experimental block), that the instruction is given in optimal temporal intervals before the task (to enable the participants to get ready (these two papers of Langner et al. and Polzien et al. are recommendable as they highlight on this aspect, the first in more general terms (doi:10.1037/xhp0000561,) and the second in the context of sports (basketball) psychology (doi:10.1037/xap0000419).   (-3-) stroop task  The calculation of effects for the Stroop task in your study appears to deviate from the conventional methodological standards typically employed for Stroop-like (conflict) tasks. It is important to acknowledge that the task used in your study does not strictly adhere to the traditional Stroop test or stroop-like tests. A paper by Polzien et al., which I recommended earlier, nicely explains these aspects in the context of sports psychology (though they used a stroop-like paradigm known as the head-fake paradigm in basketball). The way performance measures are computed in your study resembles more of a reaction time effect solely for the incongruent condition, while the congruent condition is simply disregarded. In typical computerized Stroop test versions, participants are presented with colour words and are required to respond to the colour itself, with the word serving as the irrelevant dimension. There are two conditions: the congruent condition, where the word and colour are the same, and the incongruent condition, where the word and colour differ, leading to increased difficulty. The Stroop effect is usually calculated as the difference in reaction times and accuracy between these two conditions.To ensure alignment with established methodological standards for Stroop-like tasks, it is crucial to accurately measure and interpret the Stroop effect as a key performance indicator in your study design.  (-4-) statistics  the statistics including what is presented in tables is at parts incorrect or inadequate. To name one point, you use a t-test to compare the reaction times of only the incompatible trials of the stroop test with regard to the noise condition. A correct comparison would be to use the factors congruent, incongruent and noise in a three-factorial anova design separately on reaction time and errors, which would result in 7 statistical effects (3 main, 3 two-way interactions, 1 three-way interaction). To say the stroop effect is affected by noise would be validly, when there is a significant three-way interaction effect, indicating that the RT difference (the difference between congruent and incongruent trials is modulated by the factor noise). I suggest considering this in a potential revision of the manuscript.   (-5-) providing additional information in the method section, several aspects need more specification, to name only one among others, it would be crucial to know more about the noise manipulation with regard to duration, intensity, irrelevant speech effects, and so on. In general, I suggest presenting all information relevant to the experiments so that other researchers would potentially be able to replicate your work in a follow-up study. 

Reviewers' comments:

Reviewer's Responses to Questions

**Comments to the Author**

1. If the authors have adequately addressed your comments raised in a previous round of review and you feel that this manuscript is now acceptable for publication, you may indicate that here to bypass the “Comments to the Author” section, enter your conflict of interest statement in the “Confidential to Editor” section, and submit your "Accept" recommendation.

Reviewer #1: All comments have been addressed

Reviewer #2: All comments have been addressed

2. Is the manuscript technically sound, and do the data support the conclusions?

Reviewer #1: Yes

Reviewer #2: Yes

3. Has the statistical analysis been performed appropriately and rigorously? 

Reviewer #1: Yes

Reviewer #2: Yes

4. Have the authors made all data underlying the findings in their manuscript fully available?

Reviewer #1: Yes

Reviewer #2: Yes

5. Is the manuscript presented in an intelligible fashion and written in standard English?

Reviewer #1: Yes

Reviewer #2: No

6. Review Comments to the Author

Reviewer #1: The authors have addressed all the comments provided by all the reviewers. I have no further comments.

Reviewer #2: The paper is interesting and almost ready for publication, but as the other reviewer commented several times it needs professional language editing. Although it needs minor revisions in many pages still has a lot of errors. Below I have spotted some and make some additional comments for the authors to improve the quality of the paper.

Abstract

These distractions disrupt the attentional systems, ultimately putting the athletes' inhibition ability and performance to the test. Please rephrase the above statement. …specifically “to the test”.

Introduction

p. 6, ln.112. Participants in the Drat-throwing task. Please correct

p. 6, ln.117. challenge, which led to better performance on the Drat-throwing task. Please correct.

p. 6. Lns. 108-118. I would suggest here to read and report the nice discussion about the matching hypothesis. Hatzigeorgiadis, Zourbanos, and Theodorakis (2007) noticed that depending on parameters of the task, some self-talk cues can be more effective than others, explaining the equivocal results on the matching hypothesis. I would suggest you to read Zourbanos et al. 2013 in TSP.

p. 6, lns. 112-120. I am not convinced about this please elaborate.

Results

Are the tables formatted based on PlosOne and APA recommendations? Please see the specific instructions.

Discussion

Is there a figure inserted in the discussion? This is the first time I see a figure in the discussion section. Please delete it.

7. PLOS authors have the option to publish the peer review history of their article (what does this mean?). If published, this will include your full peer review and any attached files.

Reviewer #1: No

Reviewer #2: No

---

## [Author Response · Author response to Decision Letter 2]

31 Aug 2023

Response to reviewers

Dear Editor and Reviewers,

I wanted to express my heartfelt gratitude for taking the time to review my manuscript. Your insights and constructive feedback were invaluable, and I truly appreciate the expertise you brought to this review process.

Your suggestions were astute and have greatly improved the quality and clarity of the work. I have taken your feedback to heart and have made necessary revisions to address the concerns and recommendations you mentioned.

I recognize and appreciate the effort and dedication it takes to provide such thorough and insightful feedback. Please know that your input has not only improved this particular work but has also contributed to my growth as a researcher.

Thank you once again for your invaluable contribution. I look forward to possibly benefiting from your expertise in the future.

Sincerely,

Liu Yang & Yingchun Wang 

Additional Editor Comments

(-1-) theory 

The authors have utilised Eysenck's attention control theory of anxiety as the theoretical framework for the study. The main argument of this theory revolves around the notion that momentary (state) anxiety negatively impacts working memory functions, including executive control, phonological loop, and visual-spatial sketchpad. While this theory might be insightful and serve as a base metaphor in exploring individual differences in anxiety and related personality concepts such as neuroticism, it does not offer any specific predictions about the performance effects that could potentially be expected in the present research context. To address this limitation and enhance the theoretical impact of the study, it is recommended to provide a theoretical background that focuses on predicting the effects of the factors that are assumed to influence the outcome performance measures in the Stroop and Anti-saccade tasks. This theoretical background should address how the chosen factors, such as noise distraction and self-talk strategies, are expected to interact with attentional control and ultimately impact task performance. By incorporating a more predictive theoretical framework, the study findings and interpretations would be much better grounded and would likely contribute largely to the understanding of attentional control under noise distraction conditions. To guide you in the process of revision, I recommend a recent work of Schumann et al. which provides you with an overview of theory and relevant studies on the interplay of computational (directing, coordinating) and energetical (motivating) attentional resources to ensure performance stability (doi:10.3389/fpsyg.2022.867978), in both basic and applied (e.g., sports) psychology. 

Thank you for your suggestions regarding the theoretical foundation of this study. The reference you recommended (Schumann et al., 2022) elaborately details the impact of rest breaks on attention restoration in multitasking world. This article classifies rest, summarizes its effects on task performance, and discusses methodological issues. The explanation of phenomena through a theory is a complex and lengthy process, wherein the theory needs constant refinement. No matter it is Energetic Capacity Model, Strategic Resource Model, or other theories, most of their studies are derived from multitasking research. However, in the present study, both Experiment 1 and Experiment 2 focus on single-tasking research. The core objectives of the study are to explore the effects of noise distraction and self-talk on inhibition and their interactions, and develop Attention Control Theory (ACT). Your understanding of ACT is mostly accurate, but as mentioned, theories constantly evolve. Once a work is published, it no longer solely belongs to the author. As Roland Barthes puts it, "The death of the author is the birth of the reader." Since its inception in 2007, ACT has undergone 16 years of development (Eysenck et al., 2007). Its scope has broadened beyond just the effects of anxiety on attentional control, encompassing variables such as working memory (Angelopoulou & Drigas, 2021), mental health (Armstrong & Olatunji, 2012), affect (Pessoa, 2009). In my opinion, anxiety is one of the keys to ACT, with another key being the balance between stimulus-driven bottom-up processing and task-driven top-down processing. If the balance of attention system is disrupted, individuals might be more susceptible to external stimuli, leading to distractions and thus affecting attentional control. From the theoretical perspective, ACT can offer predictions for experimental results; from the results perspective, experimental results of the present study can further develop ACT. In conclusion, I believe ACT serves as a foundational theory for this study, but it's also essential to integrate multiple theories for interpreting the experimental results. Therefore, in the discussion section, explanations based on the Strategic Resource Model have been added (see more details in p.25).

(-2-) design to study "self-talk" vs. design to study "cuing" 

In sports psychology fields, the concept of self-talk is typically used to study real-life tasks such as tennis or basketball because in these tasks, the coordination of multiple mental operations is crucial as they need to be executed in a well-timed sequence and in the correct or optimal order. When the task is not already over-learned (or is below a certain level of automatisation), self-talk strategies are critical as they can aid coordinating action sequences. In other words, directional self-talk is equivalent to representing the points in a sequence in working memory prior to the actual execution of the action, ensuring that the intended sequence of movements is followed accurately. When the task is highly over-learned, then sole motivational self-talk is more essential (see Schumann et al., doi:10.3389/fpsyg.2022.867978, chap. 4.5) in ensuring the retrieval of specified sequences from memory (and not to miss it at some occasions). The present paper employs the concept and terminology of self-talk explicitly, while the actual experiments align more with other concepts referred to as "cueing" in the literature. The problem here is that this creates a misunderstanding as you link your experiments to the self-talk literature while the actual experiments are essentially designed as commonly done in the field of cognitive-experimental research. These studies focus on cueing effects, specifically involving directional cueing (e.g., look-away effect in the antisaccade task) and effort cueing (e.g., try-harder effect). To understand the mechanism of cueing, in particular to understand effort cuering relating to why some of the manipulations did not work in the present experimental context of your study, I would suggest one of my own work (doi:10.1007/s00426-016-0810-1), apologising for doing so, but in this work it is explained how try-harder manipulation work, what factors are critical and what aspects of performance is most indicative of increased motivation. In very brief, (1) motivational cues work best when pre-instructed (i.e., the cue try-hard is explained before the experiment to the participant), (2) when they are implemented as reminders during the task (because participants would forget the instruction over the course of a block of trials (see Schumann et al., 2022, chap. 5.4) when the instruction is only given before the experimental block), that the instruction is given in optimal temporal intervals before the task (to enable the participants to get ready (these two papers of Langner et al. and Polzien et al. are recommendable as they highlight on this aspect, the first in more general terms (doi:10.1037/xhp0000561,) and the second in the context of sports (basketball) psychology (doi:10.1037/xap0000419).

Thank you for your valuable suggestions and recommendations. After carefully reading the literature you provided me (Langner et al., 2018; Polzien et al., 2023; Schumann et al., 2022; Steinborn et al., 2017), I have a clearer understanding of the similarities and differences between motivational self-talk and the implementation of the effort instruction. Self-talk is defined as: “(a) verbalizations or statements addressed to the self; (b) multidimensional in nature; (c) having interpretive elements association with the content of statements employed; (d) is somewhat dynamic; and (e) serving at least two functions; instructional and motivational, for the athlete (Hardy, 2006),” and effort instruction is defined as a cue to instruct participants try harder in the trial (Steinborn et al., 2017). From a definitional standpoint, hence, motivational self-talk is different from effort instruction. In addition, the two also have slight differences in some other characters, such as the context teller, frequency and performance (see Table 1). It is clear that the current paper employs 'self-talk' correctly, rather than using 'cueing' as it is typically used in the field of cognitive-experimental research. However, motivational self-talk and effort instruction share many similarities and have the common goal of improving motivation and performance. This is also reflected in the similarity between the result of the present study and yours (Steinborn et al., 2017). Specifically, try-harder instruction resulted in a global speed-up of responses, while error rate is higher than stander condition in short foreperiod (FP, the interaction effect of FP × CUE is significant), and motivational self-talk also increased the error rate in antisaccade task. Notably, the short FP and the interval between “ready” (the signal of starting self-talk) and fixation point (the signal of starting of the antisaccade task) are both 1000ms. Therefore, we assume that the results of present study partially confirm your conclusion from another aspect, which is effort instruction in short FP hampered the precise control of the motor-system components that are decisive for performance. Finally, thank you for your constructive suggestions and recommendations again, I will add relevant information in Discussion part (see more details in p.24-25).

Table 1. Characters of motivational self-talk and effort instruction

Characters Motivational self-talk Effort instruction

Context Cue word(s) Cue word(s)

Context teller Participants themselves The text on screen

Timestamp Beginning of a trial Beginning of a trial

Frequency Every trial 20% of all trials

Interference Not interfere any process Not interfere any process

Performance Not necessarily improve Must improve 

(-3-) stroop task 

The calculation of effects for the Stroop task in your study appears to deviate from the conventional methodological standards typically employed for Stroop-like (conflict) tasks. It is important to acknowledge that the task used in your study does not strictly adhere to the traditional Stroop test or stroop-like tests. A paper by Polzien et al., which I recommended earlier, nicely explains these aspects in the context of sports psychology (though they used a stroop-like paradigm known as the head-fake paradigm in basketball). The way performance measures are computed in your study resembles more of a reaction time effect solely for the incongruent condition, while the congruent condition is simply disregarded. In typical computerized Stroop test versions, participants are presented with colour words and are required to respond to the colour itself, with the word serving as the irrelevant dimension. There are two conditions: the congruent condition, where the word and colour are the same, and the incongruent condition, where the word and colour differ, leading to increased difficulty. The Stroop effect is usually calculated as the difference in reaction times and accuracy between these two conditions.

To ensure alignment with established methodological standards for Stroop-like tasks, it is crucial to accurately measure and interpret the Stroop effect as a key performance indicator in your study design.

Thank you for your valuable feedback. We acknowledge that the standard Stroop effect is typically calculated by comparing reaction times and accuracy between the congruent and incongruent conditions. Regrettably, during the programming of the Stroop task, the inclusion of the congruent condition data was overlooked. I deeply regret that I cannot recalculate the data following the conventional method due to this oversight. The decision not to include the congruent condition data was motivated by several reasons: (1) Study Objective: The primary aim of Study 1 was to ascertain the effectiveness of noise distraction materials in attenuating inhibition. With this goal in mind, we believed that demonstrating a difference in accuracy or reaction time for the incongruent condition—between noisy and quiet conditions—would suffice. (2) Research Focus: While Study 2 is the main thrust of our research, Study 1 served primarily as a foundational stage for it. (3) Literature Precedent: Our method was influenced by Alimohammadi and Ebrahimi (2017). In their work, the effects of low-frequency and high-frequency noises on inhibitory performance were compared. Here, the key dependent variables were "incorrect reactions" and "working time" in the incongruent conditions of the Stroop task, and they did not calculate the Stroop effect as the conventional procedure. To address this methodological limitation, we will explicitly mention it in our manuscript. This ensures readers fully grasp the data constraints when interpreting our results. We appreciate the keen insights provided and hope to make the necessary amendments to enhance our paper's clarity and rigor (see more details in p.28).

(-4-) statistics 

the statistics including what is presented in tables is at parts incorrect or inadequate. To name one point, you use a t-test to compare the reaction times of only the incompatible trials of the stroop test with regard to the noise condition. A correct comparison would be to use the factors congruent, incongruent and noise in a three-factorial anova design separately on reaction time and errors, which would result in 7 statistical effects (3 main, 3 two-way interactions, 1 three-way interaction). To say the stroop effect is affected by noise would be validly, when there is a significant three-way interaction effect, indicating that the RT difference (the difference between congruent and incongruent trials is modulated by the factor noise). I suggest considering this in a potential revision of the manuscript. 

Thank you for your suggestion. We deeply regret and acknowledge an oversight in our experimental design. Due to this oversight, only data from the incongruent condition were included in Study 1. This means we cannot conduct ANOVA analysis as you suggested, and we deeply apologize for it. In our future studies, we will ensure rigorous experimental design checks to avoid such oversights. We believe in the importance of this line of inquiry and are committed to providing a comprehensive understanding of the topic. Thank you again for pointing out this crucial aspect. We greatly appreciate your patience and understanding.

(-5-) providing additional information

in the method section, several aspects need more specification, to name only one among others, it would be crucial to know more about the noise manipulation with regard to duration, intensity, irrelevant speech effects, and so on. In general, I suggest presenting all information relevant to the experiments so that other researchers would potentially be able to replicate your work in a follow-up study.

Thank you for highlighting the need for more specificity in our method section. We have expanded on the details of the noise manipulation, providing information on duration, intensity, and other relevant properties to make sure that other researchers would be able to replicate our work in the future (see more details in p.10).

References

Alimohammadi, I., & Ebrahimi, H. (2017). Comparison between effects of low and high frequency noise on mental performance. Applied Acoustics, 126, 131-135-135. https://doi.org/10.1016/j.apacoust.2017.05.021

Angelopoulou, E., & Drigas, A. (2021). Working memory, attention and their relationship: A theoretical overview. Research, Society and Development, 10(5), e46410515288. https://doi.org/10.33448/rsd-v10i5.15288

Armstrong, T., & Olatunji, B. O. (2012). Eye tracking of attention in the affective disorders: A meta-analytic review and synthesis. Clinical Psychology Review, 32(8), 704-723. https://doi.org/10.1016/j.cpr.2012.09.004

Eysenck, M. W., Derakshan, N., Santos, R., & Calvo, M. G. (2007). Anxiety and cognitive performance: attentional control theory. Emotion, 7(2), 336. https://doi.org/10.1080/02699931.2015.1081494

Hardy, J. (2006). Speaking clearly: A critical review of the self-talk literature. Psychology of Sport and Exercise, 7(1), 81-97. 

Langner, R., Steinborn, M. B., Eickhoff, S. B., & Huestegge, L. (2018). When specific action biases meet nonspecific preparation: Event repetition modulates the variable-foreperiod effect. Journal of Experimental Psychology: Human Perception and Performance, 44(9), 1313-1323. https://doi.org/10.1037/xhp0000561

Pessoa, L. (2009). How do emotion and motivation direct executive control? Trends in Cognitive Sciences, 13(4), 160-166. https://doi.org/10.1016/j.tics.2009.01.006

Polzien, A., Güldenpenning, I., & Weigelt, M. (2023). Repeating head fakes in basketball: Temporal aspects affect the congruency sequence effect and the size of the head-fake effect. Journal of Experimental Psychology: Applied, 29(2), 292-301. https://doi.org/10.1037/xap0000419

Schumann, F., Steinborn, M. B., Kürten, J., Cao, L., Händel, B. F., & Huestegge, L. (2022). Restoration of Attention by Rest in a Multitasking World: Theory, Methodology, and Empirical Evidence. Frontiers in Psychology, 13. https://doi.org/10.3389/fpsyg.2022.867978

Steinborn, M. B., Langner, R., & Huestegge, L. (2017). Mobilizing cognition for speeded action: try-harder instructions promote motivated readiness in the constant-foreperiod paradigm. Psychological research, 81(6), 1135-1151. https://doi.org/10.1007/s00426-016-0810-1

Reviewer #1: The authors have addressed all the comments provided by all the reviewers. I have no further comments.

Reviewer #2: The paper is interesting and almost ready for publication, but as the other reviewer commented several times it needs professional language editing. Although it needs minor revisions in many pages still has a lot of errors. Below I have spotted some and make some additional comments for the authors to improve the quality of the paper.

Abstract

These distractions disrupt the attentional systems, ultimately putting the athletes' inhibition ability and performance to the test. Please rephrase the above statement. …specifically “to the test”.

Thank you for your suggestion. Based on your feedback, I propose the following revision: "These distractions disrupt the attentional systems, ultimately compromise the athletes' inhibition ability and directly affect their performance on the court."

Introduction

p. 6, ln.112. Participants in the Drat-throwing task. Please correct

Thank you for your suggestion. I apologize for the typo. This mistake has been corrected in the revised manuscript.

p. 6, ln.117. challenge, which led to better performance on the Drat-throwing task. Please correct.

Thank you for your suggestion. I apologize for the typo. This mistake has been corrected in the revised manuscript.

p. 6. Lns. 108-118. I would suggest here to read and report the nice discussion about the matching hypothesis. Hatzigeorgiadis, Zourbanos, and Theodorakis (2007) noticed that depending on parameters of the task, some self-talk cues can be more effective than others, explaining the equivocal results on the matching hypothesis. I would suggest you to read Zourbanos et al. 2013 in TSP.

Thank you for your suggestion. After reviewing the mentioned paper (Hatzigeorgiadis et al., 2007; Zourbanos et al., 2013), I found their insights further complemented my understanding, and I have referenced their work where relevant to strengthen the introduction in my manuscript (see more details in p.6, lns104-117). 

p. 6, lns. 112-120. I am not convinced about this please elaborate. 

Thank you for your suggestion. We have expanded on this by providing a more details, and we believe these additions provide a more comprehensive understanding of self-talk (see more details in p.7, lns 117-146).

Results

Are the tables formatted based on PlosOne and APA recommendations? Please see the specific instructions.

Thank you for your suggestion. The formal of tables follows “PLOS Manuscript Body Formatting Guidelines”.

Discussion

Is there a figure inserted in the discussion? This is the first time I see a figure in the discussion section. Please delete it. 

Thank you for your question and suggestion. There indeed is a figure inserted in the discussion, aiming to visually represent the ND-ST-AC Integration Model (ND: noise distraction; ST: self-talk; AC: attentional control). We believe this figure greatly aids readers in grasping the interactions under discussion. Having a visual representation within the discussion can streamline the understanding of our proposed model and its implications. This approach, while perhaps less traditional, is not without precedent. For instance, Feher da Silva and Hare (2020) in their article published in Nature Human Behaviour inserted a figure in the discussion to clarify differences in the interpretation of results by two theories. Similarly, Shirer et al. (2012) also included a supplementary figure in their discussion section to elucidate anticipated improvements in imaging techniques as technology evolved. Therefore, given the figure's significance in clarifying the model's intricacies and aiding readers' understanding, we humbly request to retain it in the discussion section.

Feher da Silva, C., & Hare, T. A. (2020). Humans primarily use model-based inference in the two-stage task. Nature Human Behaviour, 4(10), 1053-1066. https://doi.org/10.1038/s41562-020-0905-y

Hatzigeorgiadis, A., Zourbanos, N., & Theodorakis, Y. (2007). The Moderating Effects of Self-Talk Content on Self-Talk Functions. Journal of Applied Sport Psychology, 19(2), 240-251. https://doi.org/10.1080/10413200701230621

Shirer, W. R., Ryali, S., Rykhlevskaia, E., Menon, V., & Greicius, M. D. (2012). Decoding Subject-Driven Cognitive States with Whole-Brain Connectivity Patterns. Cerebral Cortex, 22(1), 158-165. https://doi.org/10.1093/cercor/bhr099

Zourbanos, N., Hatzigeorgiadis, A., Bardas, D., & Theodorakis, Y. (2013). The Effects of Self-Talk on Dominant and Nondominant Arm Performance on a Handball Task in Primary Physical Education Students. The Sport Psychologist, 27(2), 171-176. https://doi.org/10.1123/tsp.27.2.171

---

## [Editor Report · Decision Letter 3]

18 Sep 2023

The effect of motivational and instructional self-talk on attentional control under noise distraction

PONE-D-22-31246R3

Dear Dr. Wang,

We’re pleased to inform you that your manuscript has been judged scientifically suitable for publication and will be formally accepted for publication once it meets all outstanding technical requirements.

Editor comments: The manuscript has undergone considerable improvement, facilitated by an extensive academic dialogue among scholars (i.e., between authors, reviewers, and editor) from slightly differing disciplines. This discourse has effectively bridged the gap between basic cognitive and applied sport psychology, constituting a significant strength of this work. While there remain some weaknesses (e.g., the implemention of stroop paradigm), the authors have deliberately addressed these aspects within the manuscript. Consequently, I find the current submission suitable for publication and am convinced it will make a substantive contribution to the relevant fields.

Kind regards,

Michael B. Steinborn, PhD

Section Editor

PLOS ONE
---

## [Editor Report · Acceptance letter]

20 Sep 2023

PONE-D-22-31246R3 

The effect of motivational and instructional self-talk on attentional control under noise distraction 

Dear Dr. Wang:

I'm pleased to inform you that your manuscript has been deemed suitable for publication in PLOS ONE. Congratulations! Your manuscript is now with our production department. 

Kind regards, 

on behalf of

Dr. Michael B. Steinborn 

Section Editor

PLOS ONE